# LATENT-SPACE DENOISING FOR CAUSAL REPRESENTATION LEARNING VIA FREE-ENERGY-GUIDED WASSERSTEIN PARTICLE FLOWS

## ABSTRACT

Learning from corrupted observations is ubiquitous in practice, yet standard training procedures often fail under unknown nonlinear mixing and realistic noise. In causal representation learning (CRL), estimates of latent factors and their causal structure are particularly brittle to such mixing effects. We address this by denoising in a learned latent space, where the corruption approximately follows an additive noise model realized via an embedding encoder. We recover the clean latent distribution by minimizing a free-energy objective function, which couples a Kullback–Leibler divergence between the convolved clean model and the observed embedding distribution with an entropy regularizer for stability. From this objective function, we further compute the variational derivatives, derive a weighted Wasserstein gradient, and design an explicit particle flow algorithm to carry out the latent-space denoising. The resulting denoiser functions as a drop-in module for CRL and, across noisy real-world and simulated datasets, improves overall accuracy and structural recovery relative to standard CRL baselines.

## 1 INTRODUCTION

Uncovering the latent factors that govern complex systems has deep roots in signal processing (Comon, 1994; Hyvärinen & Pajunen, 1999), representation learning (Bengio et al., 2013; LeCun et al., 2015), and causal inference (Pearl, 2009; Peters et al., 2017). A central ambition of modern machine learning is to infer a causal latent ground-truth model from low-level observations and leverage it for robust prediction, explanation, and control. Causal representation learning (CRL) (Schölkopf et al., 2021) crystallizes this agenda by seeking learned embeddings aligned with the underlying generative mechanisms, promising the application of causal reasoning to unstructured high-dimensional data, improved out-of-distribution generalization, and the construction of mechanistic world models. Recent advances span identifiability under unknown single- and multi-node interventions (Buchholz et al., 2023; Varıcı et al., 2024), multi-environment identifiability (Jin & Syrgkanis, 2024), temporal formulations (Lippe et al., 2022; Lachapelle et al., 2024; Song et al., 2024; Li et al., 2025), invariance-based principles for CRL (Yao et al., 2025), and score-based approaches for linear and general transformations(Varıcı et al., 2025).

Despite progress on identifiability with added structure (e.g., interventions or environment/view/time information), current CRL methods remain brittle on real data. Empirical sanity checks built on controlled, ground-truthed systems (Gamella et al., 2025a;b) show that representative approaches—interventional (Buchholz et al., 2023), multiview (Yao et al., 2024), and temporal ((Lippe et al., 2022))—fail to recover latent factors under realistic sensor noise and nonlinear mixing, with failures persisting even on favorable deterministic datasets. The central barrier to deploying CRL is therefore not theoretical identifiability, but estimation fragility under realistic conditions.

To address the practical fragility of CRL, we propose a latent-space denoising method that directly learns the clean latent distribution to correct corrupted latent variables. We employ an embedding map under which the corruption is approximately additive, and cast distribution recovery as the minimization of a free-energy functional that balances a Kullback–Leibler (KL) divergence between the Gaussian-convolved clean model and the observed embedding distribution against a negative Shannon-entropy regularizer. Minimizing this objective yields a distributional denoiser that can be

easily plugged into existing CRL pipelines without architectural modifications. When attached to CRL baselines, our denoiser substantially improves performance on noisy real-world (and simulated) datasets, enhancing both factor estimation and structural recovery. The superiority of our proposed method over contrastive CRL (CCRL) (Buchholz et al., 2023; Gamella et al., 2025a) is demonstrated in Table 1.

| Method | Real-World | | | Synthetic (Zero Noise) | | | Synthetic (Noise = 0.002)[1] | | |
|---|---|---|---|---|---|---|---|---|---|
| | MCC↑ | SHD↓ | AUROC↑ | MCC↑ | SHD↓ | AUROC↑ | MCC↑ | SHD↓ | AUROC↑ |
| CCRL | $0.30 \pm 0.02$ | $3.00 \pm 0.71$ | $0.80 \pm 0.11$ | $0.88 \pm 0.03$ | $1.00 \pm 0.71$ | $0.92 \pm 0.05$ | $0.50 \pm 0.05$ | $2.60 \pm 0.55$ | $0.77 \pm 0.03$ |
| Ours | $\mathbf{0.54 \pm 0.05}$ | $\mathbf{2.00 \pm 0.71}$ | $\mathbf{0.84 \pm 0.08}$ | $\mathbf{0.97 \pm 0.00}$ | $\mathbf{0.40 \pm 0.55}$ | $\mathbf{0.98 \pm 0.04}$ | $\mathbf{0.64 \pm 0.04}$ | $\mathbf{2.20 \pm 0.45}$ | $\mathbf{0.85 \pm 0.07}$ |

Table 1: Comparison of CCRL and our proposed method across metrics on real-world and synthetic datasets. The experimental results demonstrate that our proposed method significantly outperforms the CCRL baseline across all metrics on all three datasets: real-world (lt_crl_benchmark_v1), zero-noise synthetic, and high-noise synthetic. Our method yields substantially higher latent-factor recovery (MCC), and for causal structural discovery it achieves lower Structural Hamming Distance (SHD) and higher area under the ROC curve (AUROC).

Specifically, we model observations in a low-dimensional latent domain where an encoder maps data into a space in which corruption behaves approximately as additive noise; in this space, predicting the observed embeddings corresponds to applying a convolution operator to a clean latent distribution. Under mild noise conditions (e.g., nowhere-vanishing characteristic function), the clean signal is theoretically identifiable via deconvolution. Building on this, our method does not aim to learn a better embedding, but instead recovers the clean latent distribution directly. We introduce a free-energy objective that balances two forces: a Kullback–Leibler fit between the smoothed model and the observed embedding distribution, and a negative Shannon-entropy regularizer for stability. A variational analysis of this objective yields a principled steepest-descent dynamics in a weighted Wasserstein geometry with a strict energy-dissipation property. For computation, we instantiate this dynamics as an explicit particle-flow procedure: the latent distribution is represented by particles that are iteratively transported using a smoothed score-mismatch term together with the entropy regularizer; the observed embedding scores are obtained via kernel density estimation.

Our free-energy objective admits a variational calculus that yields a mobility-weighted Wasserstein gradient and, in turn, a particle-flow scheme that transports empirical latent samples toward the recovered clean distribution. Casting KL descent as a gradient flow in Wasserstein space provides a principled bridge between optimization, partial differential equations, and sampling: the classical minimizing-movement scheme of Jordan et al. (1998) (JKO) shows that such dynamics are gradient flows respect to the $W_2$ metric, a viewpoint formalized on metric-measure spaces by Ambrosio et al. (2008). On the algorithmic side, particle-based variational inference implements KL descent with finite particles and often attains favorable sample efficiency (Liu & Wang, 2016; Chen et al., 2018; Dong et al., 2023). Within this function-space family, methods choose a vector field from a hypothesis class (e.g., a reproduced kernel Hilbert space or a preconditioned function class) and deterministically push particles to reduce KL, but performance can be sensitive to kernel/metric design; Cheng et al. (2023) mitigate this by recasting the particle-based variational inference as a generalized Wasserstein gradient flow with adaptive metrics. In contrast, Wibisono (2018) treats sampling as optimization over probability measures: KL descent is realized as the Wasserstein-2 gradient flow on the space of finite-second-moment probability measures, whose continuum limit recovers Langevin dynamics; this measure-space perspective makes energy dissipation explicit and motivates JKO-type discretizations, without appealing to variational inference or parametric-family constraints. By comparison, Lambert et al. (2022) recast variational inference itself as a Wasserstein gradient flow constrained to Gaussian and Gaussian-mixture families, working on the Bures–Wasserstein manifold of Gaussian measures (a Riemannian submanifold of the space of finite-second-moment probability measures), which yields a tractable stochastic gradient decent algorithm with non-asymptotic convergence guarantee. Our approach also builds on the measure-space gradient-flow calculus, but targets a different problem and objective: we neither perform variational inference nor restrict ourselves to Gaussian families. Rather than performing posterior sampling, we seek to recover a clean latent distribution from corrupted embeddings by minimizing a free-energy objective tailored to latent-space denoising, and follow a mobility-weighted Wasserstein geometry with explicit particle-flow updates aligned with the forward convolution operator.

---

[1]Defined in scale $[0, 1/255]$ corresponding to a standard deviation of $2 * 10^{-3} * 255^2$ in 8-bit pixel units.

Our free-energy-guided Wasserstein denoiser plugs into existing CRL pipelines to denoise and refine the latent distribution, providing cleaner inputs for downstream causal learners. Our contributions are threefold: (i) Formulation—we cast latent-space denoising as minimization of a free-energy functional objective that balances a KL fit between the convolved clean model and the observed embedding distribution with a (negative) Shannon-entropy regularizer; (ii) Geometry-to-algorithm—we derive the mobility-weighted Wasserstein gradient (steepest-descent vector field) of this objective and instantiate it as an explicit particle-flow algorithm that tracks the measure-space gradient flow; and (iii) Plug-in effectiveness—inserted as a drop-in step, our denoiser improves factor estimation and structural recovery on noisy real-world datasets where prior pipelines struggled (Gamella et al., 2025a), and strengthens CRL baselines on both real and simulated data.

## 2   RELATED WORK

**On gradient flow in Wasserstein space.**   A long line of work focuses on inference and sampling as steepest descent of an energy over the Wasserstein space of probability measures, with the Jordan–Kinderlehrer–Otto scheme and metric-space gradient-flow theory providing the backbone (Jordan et al., 1998; Ambrosio et al., 2008; Otto, 2001; Santambrogio, 2015; Peyré & Cuturi, 2019). Within this measure-space calculus, KL descent recovers Fokker–Planck/Langevin dynamics and admits proximal discretizations, which has inspired both samplers and variational methods. Recent results specialize this geometry to variational inference on tractable manifolds: Lambert et al. (2022) identify Gaussian (and Gaussian-mixture) variational inference as Bures–Wasserstein gradient flows and develop stochastic gradient descent algorithm with non-asymptotic guarantee. In parallel, particle-based variational inference realizes KL descent with finite particle systems: kernelized method (Liu & Wang, 2016) chooses a vector field in a reproducing kernel Hilbert space to transport particles, while generalized Wasserstein formulations introduce adaptive metrics and strengthened convergence behavior (Cheng et al., 2023; Duncan et al., 2023).

**On causal representation learning.**   Causal representation learning (CRL) seeks to recover latent causal factors from high-dimensional observations (Schölkopf et al., 2021). Interventions play a central role in achieving identifiability: single-node interventions suffice under nonlinear mixing (Buchholz et al., 2023). Furthermore, the identifiability guarantees can be extended through interventional distributions (Ahuja et al., 2023), causal component analysis with known graphs (Liang et al., 2023), and linear disentanglement where one intervention per variable is necessary and sufficient (Squires et al., 2023), accompanied by practical implementations including iVAE (Khemakhem et al., 2020) and denoising-based DVAE methods (Im et al., 2017). More recent advances have relaxed the restrictive single-node assumption, proving identifiability from multi-node interventions (Bing et al., 2024) and even in fully nonparametric settings with unknown interventions (von Kügelgen et al., 2023). Alongside these developments, broader perspectives have emerged, including temporal approaches such as CITRIS (Lippe et al., 2022), and mechanism-sparsity formulations (Lachapelle et al., 2022; 2024). Despite this progress, real-world evaluations remain limited; recent light-tunnel experiments have revealed severe brittleness of CRL pipelines under realistic noise and imperfect mixing (Gamella et al., 2025a;b).

## 3   LATENT-SPACE DENOISING VIA WASSERSTEIN PARTICLE FLOWS

### 3.1   OBSERVATION MODEL AND LOW-DIMENSIONAL EMBEDDING

Let $Z \in \mathbb{R}^{d_z}$ be a latent random variable and let $f : \mathbb{R}^{d_z} \to \mathbb{R}^{d_x}$ be a signal mixing map. Define the clean signal $X = f(Z) \in \mathbb{R}^{d_x}$. The observation $Y$ follows an additive-noise model

$$Y \;=\; X + \varepsilon, \tag{1}$$

where the noise $\varepsilon$ is independent of $X$. We write $x$ and $y$ for realizations of $X$ and $Y$, respectively.

The following proposition shows that the clean signal $X$ is identifiable from $Y$ when the noise has an infinitely divisible distribution. Related deconvolution-based identifiability results for more complex time-series models with Gaussian noise—where the noise variance is treated as a learnable scalar—are discussed in Lachapelle et al. (2022; 2024). The key tool is the nonvanishing of characteristic functions; these works specialize it to the Gaussian case.

**Proposition 1** (Identifiability of the noiseless signal). *Consider the additive noise model $Y = X + \varepsilon$ in (1). If the noise $\varepsilon$ has a non-vanishing characteristic function, then the distribution of $Y$ uniquely determines the distribution of $X$. In particular, this condition holds whenever $\varepsilon$ is infinitely divisible, hence $X$ is identifiable in that case.*

Infinitely divisible noise has a nonvanishing characteristic function and thus guarantees identifiability in the additive model. The class of infinitely divisible laws includes Gaussian, Cauchy, Gamma, Laplace, Poisson, and negative binomial distributions. Identifiability results in (Buchholz et al., 2023) are stated for a noiseless nonlinear mixing model $X = f(Z)$ under interventions. In our additive-noise setting $Y = X + \varepsilon$ with $X \perp \varepsilon$, Proposition 1 implies that the distribution of $Y$ uniquely determines the distribution of $f(Z)$ by deconvolution. Consequently, the noisy case reduces to the noiseless one treated by Buchholz et al. (2023), and their identifiability guarantees (up to permutation and scaling) continue to hold under additive noise.

We introduce an encoder $h : \mathbb{R}^{d_x} \to \mathbb{R}^{d_u}$ to approximately invert the mixing map $f$ and work with a low-dimensional embedding since typically $d_x > d_u$. For a probability measure $P$ on $\mathbb{R}^{d_x}$ and measurable $h$, the pushforward $h_{\#}P$ (Figalli & Glaudo, 2021) is defined by $(h_{\#}P)(A) = P(h^{-1}(A))$ for all Borel $A \subset \mathbb{R}^{d_u}$. Define the clean and observed embeddings $V = h(X)$ and $U = h(Y)$. Write their laws as $\mu_V := \mathrm{Law}(V) = h_{\#}\mathrm{Law}(X), \mu_U := \mathrm{Law}(U) = h_{\#}\mathrm{Law}(Y)$. Assume $\mu_V$ and $\mu_U$ are absolutely continuous w.r.t. Lebesgue measure, and denote their densities by $\rho$ and $\nu$ so that $\mu_V = \rho\, du$ and $\mu_U = \nu\, du$. The encoder reduces the ambient dimension, making subsequent nonparametric estimation computationally feasible and statistically stable; this motivates the conservative rule $d_x \geq d_u \geq d_z$, and when $d_z$ is known we set $d_u = d_z$ by default.

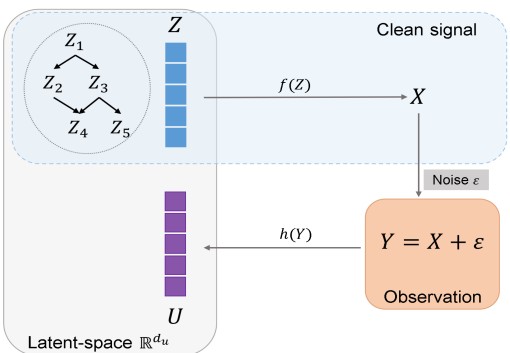

Figure 1: Latent-space observation model. The encoder $h : \mathbb{R}^{d_x} \to \mathbb{R}^{d_u}$ approximates the inverse of the nonlinear mixing map $f$, embedding high-dimensional observations into a low-dimensional latent space. This construction yields observed embeddings $U = h(Y)$.

Using a first-order Taylor expansion of $h$ at $X$, we obtain $U = h(X + \varepsilon) \approx h(X) + J_h(X)\varepsilon$, where $J_h(X)$ is the Jacobian of $h$ at $X$. If the mixing map $f : \mathbb{R}^{d_z} \to \mathbb{R}^{d_x}$ is injective and the encoder is chosen (or trained) so that $h \simeq f^{-1}$, then the clean embedding satisfies $V = h(X) \approx Z$ and thus $\mu_V \simeq \mathrm{Law}(Z)$. Consequently, the observed embedding obeys the local latent additive-noise model $U \approx Z + J_h(X)\varepsilon$, i.e., after linearization the noise remains additive without any Gaussianity assumption on $\varepsilon$. Under a local "frozen Jacobian" assumption (i.e., $J_h$ is nearly constant on the region of interest), the observed embedding law $\mu_U$ is approximated by a convolution of the clean embedding law $\mu_V$ with the noise kernel generated by the local linearization.

### 3.2 Objective Function for Latent Space Denoising

As in the previous subsection, the encoder $h : \mathbb{R}^{d_x} \to \mathbb{R}^{d_u}$ induces clean and observed latent variables $V = h(X)$ and $U = h(Y)$ with densities $\rho$ and $\nu$ on a low-dimensional latent space. We work on a domain $\Omega \subseteq \mathbb{R}^{d_u}$ and write $\mathcal{P}_2(\Omega) := \{ \rho \in L^1(\Omega) : \rho \geq 0,\ \int_\Omega \rho\, du = 1,\ \int_\Omega |u|^2 \rho(u)\, du < \infty \}$ for the set of probability densities on $\Omega$ with finite second moment (so $\rho, \nu \in \mathcal{P}_2(\Omega)$). The mobility weighted Wasserstein structure that endows $\mathcal{P}_2(\Omega)$ with a local norm will be introduced in the next subsection.

Given a candidate clean embedding density $\rho \in \mathcal{P}_2(\Omega)$, the additive-noise model motivates the smoothed (convolved) density $q_\rho := \rho * \varphi_H$, where $\varphi_H$ is the centered Gaussian kernel on $\mathbb{R}^{d_u}$ with covariance $H \succ 0$. We then define a free-energy objective $\mathcal{E} : \mathcal{P}_2(\Omega) \to \mathbb{R}$ combining a data-fit term (KL divergence in the embedding space) with an entropy regularizer:

$$\mathcal{E}(\rho) = \underbrace{D_{\mathrm{KL}}(q_\rho \,\|\, \nu)}_{\text{predicted vs. observed in latent space}} + \lambda \underbrace{\mathrm{Ent}(\rho)}_{\text{negative Shannon entropy}}, \qquad \lambda \geq 0, \qquad (2)$$

where $D_{\mathrm{KL}}(p\|q) = \int p(u)\log\big(p(u)/q(u)\big)\,du$ and $\mathrm{Ent}(\rho) = \int \rho(u)\log\rho(u)\,du$. Since the Shannon entropy is $\mathrm{H}(\rho) = -\mathrm{Ent}(\rho)$, we may equivalently write $\mathcal{E}(\rho) = D_{\mathrm{KL}}(q_\rho \,\|\, \nu) - \lambda\,\mathrm{H}(\rho)$, which parallels the usual free-energy form "internal energy minus temperature times entropy."

Minimizing $\mathcal{E}(\rho)$ seeks a deblurred latent density $\rho$ whose Gaussian blur $q_\rho$ best matches the observed embedding $\nu$ under the additive-noise channel. The KL term measures the mismatch between the model prediction $q_\rho$ and the data $\nu$, discouraging mass where $\nu$ is small and aligning $q_\rho$ with the observed distribution. The entropy term serves as a maximum-entropy regularizer: it suppresses spurious spikes and mode collapse, promotes smoothness, and stabilizes the deconvolution. The parameter $\lambda$ controls the fit-robustness tradeoff: smaller $\lambda$ yields sharper but noise-sensitive reconstructions, whereas larger $\lambda$ favors more spread-out, stable estimates.

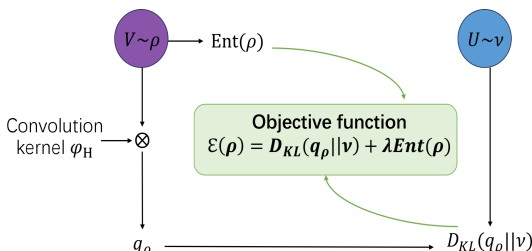

Figure 2: Objective function for denoising. The energy $\mathcal{E}(\rho) = D_{\mathrm{KL}}(q_\rho \,\|\, \nu) + \lambda\,\mathrm{Ent}(\rho)$ combines a KL divergence term, which aligns the convolution density $q_\rho = \rho * \varphi_H$ with the observed embedding distribution $\nu$, and an entropy regularizer.

We now compute the first variation of the population objective that governs the latent-space dynamics. The following theorem collects the required first variations and their spatial gradients, which are used to derive the mobility weighted Wasserstein gradient update in latent space.

**Lemma 1** (First variation and its gradient). *Assume $\rho, \nu : \mathbb{R}^{d_u} \to (0, \infty)$ are $C^1$ probability densities and $\nabla\log\rho, \nabla\log\nu$ are continuous and bounded on every compact set. Let $\varphi_H$ be a centered Gaussian with covariance $H \succ 0$ and set $q_\rho = \rho * \varphi_H$. Assume $\mathrm{Ent}(\rho) < \infty$ and $D_{\mathrm{KL}}(q_\rho \| \nu) < \infty$. The first variation of the objective function (2) is*

$$\frac{\delta}{\delta\rho}\mathcal{E}(\rho) = \underbrace{\big(\varphi_H * \log\tfrac{q_\rho}{\nu}\big)(u)}_{\textit{from } \mathrm{KL}(q_\rho\|\nu)} + \lambda\,\underbrace{\log\rho(u)}_{\textit{from } \mathrm{Ent}(\rho)}. \tag{3}$$

*Moreover the spatial gradient of the first variation $\psi = \frac{\delta}{\delta\rho}\mathcal{E}$ is*

$$\nabla\psi(u) = \underbrace{\big(\varphi_H * (\nabla\log q_\rho - \nabla\log\nu)\big)(u)}_{\textit{smoothed score mismatch}} + \lambda\,\nabla\log\rho(u). \tag{4}$$

Equation (3) shows that the first variation splits into a smoothed log-density ratio (from the $D_{\mathrm{KL}}$ term) and the entropy score (from $\mathrm{Ent}$), and equation (4) provides the corresponding spatial gradient.

### 3.3 Wasserstein Riemannian gradient and its induced particle flow update

Let $\Sigma : \Omega \to \mathbb{R}^{d_u \times d_u}$ be a smooth, symmetric, positive-definite mobility. For $\rho \in \mathcal{P}_2(\Omega)$ define the $\Sigma$-weighted velocity space $L^2_{\rho, \Sigma^{-1}}(\Omega; \mathbb{R}^{d_u}) = \{w : \int_\Omega w^\top \Sigma^{-1} w\rho du < \infty\}$ and the pre-tangent set $T_\rho\mathcal{P}_2(\Omega) = \{-\nabla\cdot(\rho w) : w \in L^2_{\rho, \Sigma^{-1}}\}$ (Appendix A.3). For $\sigma = -\nabla\cdot(\rho w)$ we define the $\Sigma$-weighted norm by $\inf_{-\nabla\cdot(\rho w)=\sigma} \int_\Omega w^\top \Sigma^{-1} w\,\rho\,du$, whose unique minimizer is $w^* = \Sigma\nabla\phi$ solving $-\nabla\cdot(\rho\,\Sigma\nabla\phi) = \sigma$ (with $\phi$ up to a constant). Hence $\|\sigma\|^2_{\rho, \Sigma} = \int_\Omega \nabla\phi^\top \Sigma\nabla\phi\,\rho\,du$, yielding a local $\Sigma$-weighted Wasserstein (Riemannian) structure on $\mathcal{P}_2(\Omega)$; for $\sigma_i = -\nabla\cdot(\rho\,\Sigma\nabla\phi_i)$ we set $\langle\sigma_1, \sigma_2\rangle_{\rho, \Sigma} = \int_\Omega (\nabla\phi_1)^\top \Sigma\nabla\phi_2\,\rho\,du$. The energy $\mathcal{E}$ is Gâteaux differentiable at $\rho$ if there exists $\psi = \delta\mathcal{E}/\delta\rho$ such that for every $\zeta \in C_c^\infty(\Omega)$ with $\int_\Omega \zeta\,du = 0$, $D\mathcal{E}(\rho)[\zeta] = \frac{d}{d\varepsilon}\big|_{\varepsilon=0}\mathcal{E}(\rho + \varepsilon\zeta) = \int_\Omega \psi\,\zeta\,du$. By the Riesz representation under $\langle\cdot, \cdot\rangle_{\rho, \Sigma}$ there is a unique $\mathrm{grad}_{W_{2,\Sigma}}\mathcal{E}(\rho) \in \overline{T_\rho\mathcal{P}_2(\Omega)}$ with $D\mathcal{E}(\rho)[\sigma] = \langle\mathrm{grad}_{W_{2,\Sigma}}\mathcal{E}(\rho), \sigma\rangle_{\rho, \Sigma}$ for all $\sigma \in T_\rho\mathcal{P}_2(\Omega)$. For any velocity field $w$, consider one of the following standing boundary assumptions: (A) test-function regime $\varphi \in C_c^\infty(\Omega)$ so boundary integrals vanish; (B) no-flux on bounded domains $(\rho w)\cdot n = 0$ on $\partial\Omega$; or (C) whole-space $\Omega = \mathbb{R}^{d_u}$ with compact support or sufficient decay of $\rho w$ so that the flux at infinity is zero.

**Theorem 1** (Steepest descent and energy dissipation). *Let $\mathcal{E} : \mathcal{P}_2(\Omega) \to \mathbb{R}$ be Gâteaux differentiable at $\rho$ with first variation $\psi = \delta\mathcal{E}/\delta\rho$ such that $\int_\Omega \rho\,\nabla\psi^\top \Sigma\nabla\psi\,du < \infty$. Under the standing boundary*

*assumptions, the $\Sigma$-weighted Wasserstein (Riemannian) gradient is*

$$\mathrm{grad}_{W_{2,\Sigma}}\mathcal{E}(\rho) \; = \; -\nabla\!\cdot\!\big(\rho\,\Sigma\nabla\psi\big) \; \in \; T_\rho\mathcal{P}_2(\Omega),$$

*and it simultaneously characterizes steepest descent and energy dissipation. Specifically, for every $\sigma \in T_\rho\mathcal{P}_2(\Omega)$, $D\mathcal{E}(\rho)[\sigma] = \langle\mathrm{grad}_{W_{2,\Sigma}}\mathcal{E}(\rho),\,\sigma\rangle_{\rho,\Sigma} \geq -\big\|\mathrm{grad}_{W_{2,\Sigma}}\mathcal{E}(\rho)\big\|_{\rho,\Sigma}\,\|\sigma\|_{\rho,\Sigma}$, with equality iff $\sigma = -\alpha\,\mathrm{grad}_{W_{2,\Sigma}}\mathcal{E}(\rho)$. In particular, among all unit directions the maximal instantaneous decrease of $\mathcal{E}$ is attained at $-\mathrm{grad}_{W_{2,\Sigma}}\mathcal{E}(\rho)$. Moreover, if $\rho_t$ solves the (negative) $\Sigma$-weighted Wasserstein gradient flow $\partial_t\rho_t = -\mathrm{grad}_{W_{2,\Sigma}}\mathcal{E}(\rho_t) = \nabla\!\cdot\!\big(\rho_t\,\Sigma\nabla\psi_t\big)$, where $\psi_t = \frac{\delta\mathcal{E}}{\delta\rho}(\rho_t)$, then along the flow the energy dissipates according to*

$$\frac{d}{dt}\,\mathcal{E}(\rho_t) = -\big\|\mathrm{grad}_{W_{2,\Sigma}}\mathcal{E}(\rho_t)\big\|^2_{\rho_t,\Sigma} = -\int_\Omega \rho_t\,\nabla\psi_t^\top\Sigma\nabla\psi_t\,du \; \leq 0,$$

*with equality iff $\nabla\psi_t = 0$ for $\rho_t$-a.e. points.*

Our Wasserstein gradient-flow viewpoint follows the classical framework: informally, given an energy functional $\mathcal{E}$ on probability laws, one may write the evolution as a (negative) $W_2$ gradient flow so that $\mathcal{E}$ is nonincreasing along trajectories. Equivalent formulations—via the $W_2$ Riesz representation, the continuity equation, or the minimizing-movement (JKO) scheme—are standard; see Ambrosio et al. (2008); Otto (2001); Jordan et al. (1998); Santambrogio (2015); Peyré & Cuturi (2019) for precise statements. In contrast to these classical settings, our model (i) operates in a learned latent space via an embedding encoder and uses a KL data-fit between a convolution prediction and the observed embedding, rather than a Wasserstein discrepancy, and (ii) employs a mobility-weighted flow with a possibly $\Sigma \succ 0$—recovering the classical case when $\Sigma \equiv I$—to enable anisotropic transport adapted to latent geometry.

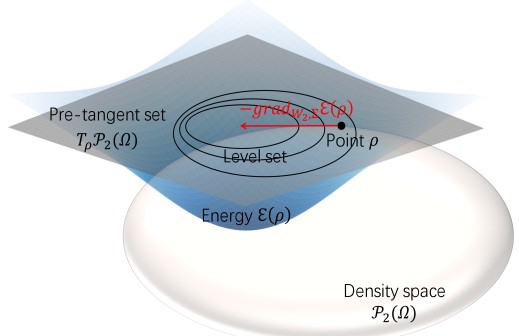

Figure 3: Geometric illustration of the Wasserstein Riemannian gradient. The blue surface represents the energy functional $\mathcal{E}(\rho)$, with its level sets shown as black lines. The white base region denotes the density space $\mathcal{P}_2(\Omega)$; the gray plane depicts the pre-tangent set $T_\rho\mathcal{P}_2(\Omega)$ at point $\rho$; and the red arrow indicates the negative gradient direction $-\mathrm{grad}_{W_2,\Sigma}\mathcal{E}(\rho)$.

We optimize an unknown density $\rho \in \mathcal{P}_2(\Omega)$ rather than a vector in Euclidean space, so updates must preserve mass and nonnegativity ($\rho \geq 0$, $\int_\Omega \rho = 1$), which suggests a kinematic particle-flow viewpoint instead of a naive Euclidean gradient step. As in Appendix B.1, we introduce a time-dependent velocity field $v_t$ and transport the density by the pushforward $\rho_t = (\Phi_t)_{\#}\rho_0$, where the flow map $\Phi_t$ solves the characteristic ODE $\dot{X}_t = v_t(X_t)$. Passing to the Eulerian description yields the continuity equation $\partial_t\rho_t + \nabla\!\cdot\!(\rho_t v_t) = 0$. Choosing the Wasserstein gradient field $v_t(u) = -\Sigma(u)\nabla\psi_t(u)$ with $\psi_t = \delta\mathcal{E}/\delta\rho(\rho_t)$ recovers the (negative) $\Sigma$-weighted Wasserstein gradient flow and the standard energy-dissipation property. Given the particle cloud $\boldsymbol{\xi}^k = \{\xi_i^k\}_{i=1}^N$ at step $k$, we update the particles using the explicit Euler scheme (Hairer et al., 1993) with step size $\tau_k > 0$:

$$\xi_i^{k+1} = \xi_i^k + \tau_k\,v\big(\xi_i^k\big) = \xi_i^k - \tau_k\,\Sigma\big(\xi_i^k\big)\,\nabla\psi^{(k)}\big(\xi_i^k\big), \qquad i = 1,\dots,N,$$

by using the steepest-descent velocity $v = -\Sigma\nabla\psi$. Using Lemma 1, we have,

$$\nabla\psi^{(k)}(u) = \big[\varphi_H * \big(\nabla\log q_{\rho^{(k)}} - \nabla\log\nu\big)\big](u) + \lambda\,\nabla\log\rho^{(k)}(u), \tag{5}$$

where $q_{\rho^{(k)}} = \rho^{(k)} * \varphi_H$. Here, the densities $\rho^{(k)}$ (from the current particle cloud $\boldsymbol{\xi}^k$) and $\nu$ (from the current mini-batch) are estimated via kernel density estimation (KDE). Define the scores $s_q^{(k)}(u) = \nabla\log q_{\rho^{(k)}}(u), s_\nu(u) = \nabla\log\nu(u), s_\rho^{(k)}(u) = \nabla\log\rho^{(k)}(u)$, and their convolution smoothings $\tilde{s}_q^{(k)} = \varphi_H * s_q^{(k)}$ and $\tilde{s}_\nu = \varphi_H * s_\nu$. Then the particles evolve via the explicit Euler step

$$\xi_i^{k+1} = \xi_i^k - \tau_k\,\Sigma(\xi_i^k)\,\big(\tilde{s}_q^{(k)}(u) - \tilde{s}_\nu(u) + \lambda\,s_\rho^{(k)}(u)\big)\Big|_{u=\xi_i^k}. \tag{6}$$

In practice, this update provides an efficient particle realization of the continuum density dynamics. Details of the particle-flow algorithm are given in Appendix B.

## 4 EXPERIMENTS

In this section, we present experimental results on both synthetic and real-world datasets to evaluate the effectiveness of our proposed method.

### 4.1 EXPERIMENT SETUP

**Implementation**   Our method is integrated as a latent space denoising module into the CCRL (Buchholz et al., 2023) and CausCell (Gao et al., 2025) implementation schemes. This module minimizes a free-energy objective that combines a KL divergence term for data fitting and an entropy regularizer for stability, trained with a mobility-weighted Wasserstein particle flow. We evaluate our method on both synthetic and real-world datasets. See Appendix C.5 for hyperparameter settings.

### 4.2 DATASETS

We evaluate on (i) two real optical-system datasets—lt_crl_benchmark_v1 and lt_camera_v1; (ii) an Optical System Synthetic Dataset; and (iii) the MERFISH Brain single-cell dataset.

**Optical System Real Datasets.**   We use the lt_crl_benchmark_v1 and lt_camera_v1 datasets (Gamella et al., 2025a;b), which involve interventions on LED brightness and polarizer angles. Each variable is intervened with 10,000 samples, yielding 60,000 images. Unlike synthetic data, these real measurements contain nontrivial noise, providing a challenging benchmark for causal representation learning. Figure 4 illustrates the ground-truth graph for lt_crl_benchmark_v1 and example frames (A–D) with their control inputs overlaid. A more detailed description of lt_camera_v1 is provided in Appendix C.2, and its experimental results are reported in Appendix C.6.

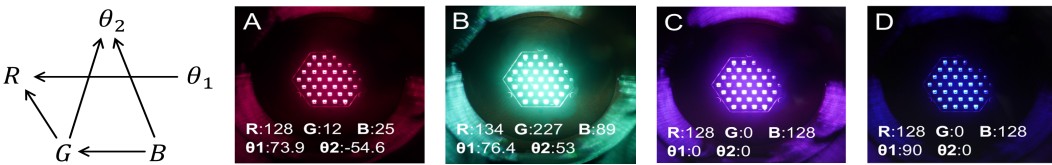

Figure 4: The ground-truth graph for the lt_crl_benchmark_v1. (A–D): Real images acquired from the dataset, with the corresponding control inputs superimposed in white. While images C and D share the same light source color $(R, G, B)$, the first polarizer in image D is rotated by $90°$, thereby illustrating the influence of the linear polarizer angles $(\theta_1, \theta_2)$.

**Optical System Synthetic Dataset (Clean and Noisy).**   We construct the optical system synthetic dataset by replacing the real light tunnel with a deterministic simulator using the same control inputs (RGB light intensities and polarizer angles). Sensor readings are taken from lt_crl_benchmark_v1 dataset, and images are generated by a multi-layer perceptron trained on an auxiliary dataset. This design removes random noise while preserving the causal structure, making the assumptions of causal representation learning more consistent and allowing controlled evaluation of algorithm performance (Gamella et al., 2025a). To generate noisy data, we inject zero-mean Gaussian noise into the latent representations of noiseless samples after encoding. The noise level is controlled by the standard deviation of the distribution: larger values cause stronger corruption in the rendered images, as shown in Figure 5. Apart from noise injection, the generation process for noisy data is identical to that of the noiseless case.

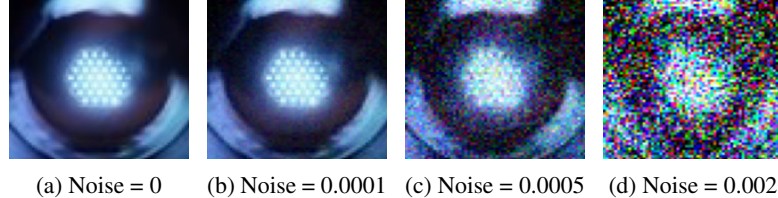

(a) Noise = 0      (b) Noise = 0.0001   (c) Noise = 0.0005   (d) Noise = 0.002

Figure 5: The effect of different noise intensities on images. The leftmost image shows the original clean version, while the others with standard deviations $10^{-4} * 255^2$, $5 * 10^{-4} * 255^2$, and $2 * 10^{-3} * 255^2$.

**MERFISH Brain Dataset.** This dataset originates from a single-cell study (Allen et al., 2023) on mouse brain aging, containing 144,249 cells and 374 genes from slice 1 of each sample. It includes four concept labels: Age (3 classes), Celltype (13 classes), Domain (8 classes), and Batch. We use the preprocessed version (Gao et al., 2025), which includes 125,672 cells. In this version, rare cell types are merged to reduce Celltype classes to 10, and Batch is excluded due to weak batch effects.

## 4.3 EVALUATION METRICS

To evaluate the recovery of causal factors and structures, we use three standard metrics: the Mean Correlation Coefficient (MCC) for alignment between learned representations and ground-truth latents, the Structural Hamming Distance (SHD) for comparing estimated and true causal graphs, and the Area Under the ROC Curve (AUROC) for classification performance (Buchholz et al., 2023; Gamella et al., 2025a). For single-cell datasets, following Gao et al. (2025), we further employ three tasks to assess reconstruction: (1) trend matching with the Pearson Correlation Coefficient (PCC) and Mean Squared Error (MSE), (2) structural consistency with Normalized Mutual Information (NMI) and the Adjusted Rand Index (ARI), and (3) fine-grained scores for marker gene preservation. Details are provided in Appendix C.3.

## 4.4 BASELINES

CCRL (Buchholz et al., 2023; Gamella et al., 2025a) establishes identifiability under nonlinear mixing. iVAE (Khemakhem et al., 2020) introduces an auxiliary-variable prior to enhance denoising, while DVAE (Im et al., 2017) couples denoising with VAEs for more robust latents. Plugging iVAE/DVAE into CCRL yields the denoising variants iCCRL and DCCRL. CausCell (Gao et al., 2025) adopts causal-representation principles for single-cell data, enabling disentanglement and controllable counterfactual generation. See Appendix C.4 for details.

## 4.5 EXPERIMENTAL RESULTS ON SYNTHETIC AND REAL DATASET

### 4.5.1 RESULTS ON OPTICAL SYSTEM SYNTHETIC DATASET

Table 2 summarizes the performance of four methods across different noise levels.

| Noise Level | Metric | CCRL | iCCRL | DCCRL | Ours |
|---|---|---|---|---|---|
| Zero | MCC↑ | $0.8776 \pm 0.0287$ | $0.4294 \pm 0.0734$ | $0.8163 \pm 0.0206$ | $\mathbf{0.9653 \pm 0.0042}$ |
| | SHD↓ | $1.0000 \pm 0.7071$ | $3.4000 \pm 0.4899$ | $2.6000 \pm 0.5477$ | $\mathbf{0.4000 \pm 0.5477}$ |
| | AUROC↑ | $0.9160 \pm 0.0477$ | $0.7480 \pm 0.0676$ | $0.8453 \pm 0.0337$ | $\mathbf{0.9773 \pm 0.0436}$ |
| 0.0001 | MCC↑ | $0.8724 \pm 0.0227$ | $0.4257 \pm 0.0745$ | $0.8306 \pm 0.0079$ | $\mathbf{0.8990 \pm 0.0130}$ |
| | SHD↓ | $1.4000 \pm 0.5477$ | $3.2000 \pm 0.7483$ | $2.6000 \pm 0.5477$ | $\mathbf{1.2000 \pm 1.0954}$ |
| | AUROC↑ | $0.8920 \pm 0.0110$ | $0.7427 \pm 0.0630$ | $0.8280 \pm 0.0513$ | $\mathbf{0.9067 \pm 0.1011}$ |
| 0.0005 | MCC↑ | $0.7610 \pm 0.0936$ | $0.4097 \pm 0.0712$ | $0.7871 \pm 0.0304$ | $\mathbf{0.8746 \pm 0.0237}$ |
| | SHD↓ | $2.4000 \pm 0.8944$ | $3.4000 \pm 0.8000$ | $2.4000 \pm 0.8944$ | $\mathbf{1.8000 \pm 1.3038}$ |
| | AUROC↑ | $0.8107 \pm 0.1022$ | $0.7027 \pm 0.0639$ | $0.8466 \pm 0.0708$ | $\mathbf{0.8547 \pm 0.1324}$ |
| 0.002 | MCC↑ | $0.4974 \pm 0.0537$ | $0.2898 \pm 0.1231$ | $\mathbf{0.7209 \pm 0.0144}$ | $0.6419 \pm 0.0358$ |
| | SHD↓ | $2.6000 \pm 0.5477$ | $3.8000 \pm 0.8367$ | $3.2000 \pm 0.4472$ | $\mathbf{2.2000 \pm 0.4472}$ |
| | AUROC↑ | $0.7733 \pm 0.0267$ | $0.6813 \pm 0.0868$ | $0.7773 \pm 0.0794$ | $\mathbf{0.8480 \pm 0.0657}$ |

Table 2: Comparison of methods across different noise levels on the optical system synthetic dataset.

In noise-free and low-noise settings, our method surpasses all baselines on MCC, SHD, and AUROC, nearly recovering the true graph. Under high noise (0.002), performance drops for all methods, yet ours remains more robust: DCCRL yields the top MCC, while we achieve the best SHD and AUROC, demonstrating robustness across noise levels.

### 4.5.2 Applications to Two Real-World Datasets

| Metric | CCRL | iCCRL | DCCRL | Ours |
|---|---|---|---|---|
| MCC↑ | $0.3016 \pm 0.0183$ | $0.4712 \pm 0.0303$ | $0.4995 \pm 0.0350$ | $\mathbf{0.5416 \pm 0.0534}$ |
| SHD↓ | $3.0000 \pm 0.7071$ | $2.2000 \pm 0.8367$ | $2.6000 \pm 0.5477$ | $\mathbf{2.0000 \pm 0.7071}$ |
| AUROC↑ | $0.8000 \pm 0.1118$ | $\mathbf{0.8387 \pm 0.0484}$ | $0.8173 \pm 0.0908$ | $0.8373 \pm 0.0787$ |

Table 3: Comparison of methods across metrics on lt_crl_benchmark_v1.

We evaluate our method on two real-world datasets: lt_crl_benchmark_v1 for causal representation learning and MERFISH Brain for biological data generation (see Tables 3 and 4; additional results appear in Appendix C.6). On the lt_crl_benchmark_v1, our method improves MCC and SHD over prior baselines, while maintaining competitive AUROC. On the MERFISH Brain, our proposed method delivers stable performance across all metrics, with notable improvements in trend matching, consistent geometric structure, and a significantly higher fine-grained score than CausCell (334.2% vs. 280.5%), which is critical for preserving marker genes.

| Method | Trend matching | | Geometric structure consistency | | Finegrained score |
|---|---|---|---|---|---|
| | PCC | MSE | ARI | NMI | Matching Score |
| CausCell | $254.0\% \pm 2.2\%$ | $32.5\% \pm 1.3\%$ | $0.3568 \pm 0.0308$ | $\mathbf{0.5834 \pm 0.0170}$ | $280.5\% \pm 26.2\%$ |
| Ours | $\mathbf{357.4\% \pm 6.3\%}$ | $\mathbf{33.4\% \pm 1.3\%}$ | $\mathbf{0.3569 \pm 0.0448}$ | $0.5750 \pm 0.0083$ | $\mathbf{334.2\% \pm 9.7\%}$ |

Table 4: Comparison of methods across metrics [2] on MERFISH Brain.

### 4.5.3 Ablation Experiment

| Metric | CCRL | Ours | w/o Entropy | w/o KL |
|---|---|---|---|---|
| MCC↑ | $0.3349 \pm 0.0250$ | $\mathbf{0.5416 \pm 0.0534}$ | $0.5248 \pm 0.0408$ | $0.3349 \pm 0.0250$ |
| SHD↓ | $2.8000 \pm 0.4472$ | $\mathbf{2.0000 \pm 0.7071}$ | $2.4000 \pm 0.8944$ | $2.8000 \pm 0.4472$ |
| AUROC↑ | $0.7880 \pm 0.0595$ | $\mathbf{0.8373 \pm 0.0787}$ | $0.8120 \pm 0.0774$ | $0.7880 \pm 0.0595$ |

Table 5: Ablations of the denoising mechanism on lt_crl_benchmark_v1.

The ablation study in Table 5 examines the roles of the KL divergence and entropy terms in our denoising mechanism. The baseline CCRL without denoising performs poorly on noisy data. The full model ("Ours") achieves the best performance. When the entropy term is omitted (w/o Entropy), the performance remains significantly better than the baseline but lower than the full model, indicating that the entropy term contributes to performance improvement, though its impact is relatively limited. In contrast, removing the KL term (w/o KL) yields performance identical to CCRL, since the entropy term alone does not drive parameter updates.

## 5 LIMITATION AND FUTURE WORK

Our denoising operates in the low-dimensional latent space, where we estimate densities and scores primarily with a standard Gaussian-kernel KDE. While this nonparametric choice is flexible, it remains sensitive to dimensionality and sample size, reflecting a bias-variance trade-off. Importantly, the Gaussian kernel is only one instantiation: the denoising module is plug-and-play and can be embedded into other CRL pipelines by swapping in alternative density/score estimators or reusing learned representations. In future work, we will validate the module across diverse CRL denoising settings (e.g., temporal formulations, varied architectures, and multiple datasets).

---

[2]The percentage represents the improvement of the experiment compared to the control trial and calculation formula is: $|(V_{\exp} - V_{\mathrm{ctrl}})/V_{\mathrm{ctrl}} \times 100\%|$. A higher percentage indicates better method performance.

## REPRODUCIBILITY STATEMENT

Proofs of all theoretical results are given in Appendix A, and algorithmic details are provided in Appendix B. Implementation details appear in Appendix C.1, dataset descriptions in Appendix C.2, and hyperparameters in Appendix C.5. Our code is included in the supplementary materials to facilitate transparency and reproducibility.

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

## A    THEORETICAL GUARANTEE

### A.1    IDENTIFIABILITY IN THE ADDITIVE NOISE MODEL

**Proposition 1** (Identifiability in the additive noise model via non-vanishing characteristic function).
*Let $Y = X + \varepsilon$ on $\mathbb{R}^d$ with $X \perp\!\!\!\perp \varepsilon$. Assume the noise has a non-vanishing characteristic function:*

$$\phi_\varepsilon(t) := \mathbb{E}\big[e^{it^\top \varepsilon}\big] \neq 0 \quad \text{for all } t \in \mathbb{R}^d.$$

*Then the law of $Y$ uniquely determines the law of $X$. Equivalently, for a fixed noise law,*

$$\mathrm{Law}(Y_1) = \mathrm{Law}(Y_2) \implies \mathrm{Law}(X_1) = \mathrm{Law}(X_2).$$

*Proof.* By independence, the characteristic function of the sum factorizes:

$$\phi_Y(t) = \mathbb{E}\Big[e^{it^\top(X+\varepsilon)}\Big] = \mathbb{E}\Big[e^{it^\top X}\Big] \mathbb{E}\Big[e^{it^\top \varepsilon}\Big] = \phi_X(t)\,\phi_\varepsilon(t) \qquad (t \in \mathbb{R}^d).$$

Since $\phi_\varepsilon(t) \neq 0$ for all $t$, we can solve $\phi_X(t) = \phi_Y(t)/\phi_\varepsilon(t)$. By the uniqueness of characteristic functions (Chung, 2001), the distribution of $X$ is uniquely determined by the distribution of $Y$.    $\square$

*Remark* 1. A sufficient condition for identifiability is that the characteristic function $\phi_\varepsilon$ is non-vanishing. This holds for any infinitely divisible (ID) law by the Lévy–Khintchine formula:

$$\phi_\varepsilon(t) = \exp\Big(im^\top t - \tfrac{1}{2}t^\top \Sigma t + \int_{\mathbb{R}^d} \big(e^{it^\top x} - 1 - it^\top x\,\mathbf{1}_{\{\|x\|<1\}}\big)\,\nu(dx)\Big) \neq 0, \qquad t \in \mathbb{R}^d.$$

Hence any ID noise has a non-vanishing characteristic function. Representative examples include the Gaussian, Laplace, Poisson, Gamma (and thus chi-square), negative binomial, and Cauchy.

By contrast, some noises have zeros in their characteristic functions and thus do not guarantee identifiability without extra information: e.g., uniform noise on $[-a, a]$ with $\phi(t) = \mathrm{sinc}(at)$ (zeros at $t = k\pi/a$), Rademacher noise with $\phi(t) = \cos t$.

*Remark* 2. The identification basing on the deconvolution is also discussed by Lachapelle et al. (2022; 2024) for more complicated time series models with Gaussian noise, whose variance is treated as a learnable scalar. This is an non-vanishing characteristic function argument specialized to the Gaussian case.

### A.2    FIRST VARIATION OF OBJECTIVE FUNCTION

In a general setting, we work on a domain $\Omega \subseteq \mathbb{R}^{d_u}$ in the latent space, and denote the set of probability densities on $\Omega$ with finite second moment by

$$\mathcal{P}_2(\Omega) := \Big\{\rho \in L^1(\Omega) : \rho \geq 0,\ \int_\Omega \rho\,du = 1,\ \int_\Omega |u|^2\rho(u)\,du < \infty\Big\}.$$

Let $\varphi_H$ denote the centered Gaussian kernel on $\mathbb{R}^{d_u}$ with covariance $H$. Given a candidate clean embedding densitiy $\rho \in \mathcal{P}_2(\Omega)$, the additive noise model provides a convolution $q_\rho = \rho * \varphi_H$. We construct a free-energy objective function $\mathcal{E} : \mathcal{P}_2(\Omega) \to \mathbb{R}$ with the Kullback–Leibler (KL) divergence and entropy regularizer:

$$\mathcal{E}(\rho) = \underbrace{\mathrm{KL}\big(q_\rho \,\|\, \nu\big)}_{\text{predicted vs. observed in embedding space}} + \lambda \underbrace{\mathrm{Ent}(\rho)}_{\text{entropy regularizer}}, \quad \lambda \geq 0. \tag{7}$$

where the entropy $\mathrm{Ent}(\rho) = \int \rho(u)\log\rho(u)\,du$ and the Kullback–Leibler divergence $\mathrm{KL}(p\|q) = \int p(u)\log\big(p(u)/q(u)\big)\,du$.

We now compute the first variation of the objective that governs the latent-space dynamics. The following lemma collects the required first variations and their spatial gradients.

**Lemma 1** (First variation of the objective function and its gradient). *Assume $\rho, \nu : \mathbb{R}^{d_u} \to (0, \infty)$ are $C^1$ probability densities and $\nabla\log\rho, \nabla\log\nu$ are continuous and bounded on every compact set.*

*Let $\varphi_H$ be a centered Gaussian with covariance $H \succ 0$ and set $q_\rho = \rho * \varphi_H$. Assume moreover that* $\mathrm{Ent}(\rho) < \infty$ *and* $D_{\mathrm{KL}}(q_\rho \| \nu) < \infty$. *The first variation of the objective function is*

$$\frac{\delta}{\delta \rho} \mathcal{E}(\rho) \; = \; \underbrace{\left( \varphi_H * \log \frac{q_\rho}{\nu} \right)(u)}_{\textit{from } \mathrm{KL}(q_\rho \| \nu)} \; + \; \lambda \underbrace{\log \rho(u)}_{\textit{from } \mathrm{Ent}(\rho)} . \tag{8}$$

*Moreover the spatial gradient of the first variation* $\psi = \frac{\delta}{\delta \rho} \mathcal{E}$ *is*

$$\nabla \psi(u) = \underbrace{\left( \varphi_H * (\nabla \log q_\rho - \nabla \log \nu) \right)(u)}_{\textit{smoothed score mismatch}} + \lambda \nabla \log \rho(u). \tag{9}$$

*Proof.* For a smooth perturbation $\xi$ with $\int \xi \, du = 0$, define $\rho_\varepsilon := \rho + \varepsilon \xi$ and $q_{\rho_\varepsilon} := \rho_\varepsilon * \varphi_H$.

**Step 1: Variation of $q_\rho$.** By linearity of convolution,

$$q_{\rho_\varepsilon} = (\rho + \varepsilon \xi) * \varphi_H = q_\rho + \varepsilon \, (\xi * \varphi_H),$$

hence $\frac{d}{d\varepsilon}\big|_{\varepsilon=0} q_{\rho_\varepsilon} = \xi * \varphi_H$.

**Step 2: Gateaux derivative of the KL term.** Set $F(\varepsilon) = \mathrm{KL}(q_{\rho_\varepsilon} \| \nu) = \int q_{\rho_\varepsilon} \log(q_{\rho_\varepsilon}/\nu) \, du$. Write $q_\varepsilon := q_{\rho_\varepsilon} = q_\rho + \varepsilon \, \delta q$ with $\delta q := \xi * \varphi_H$ from Step 1. Using $\frac{d}{dx}(x \log x) = 1 + \log x$ and dominated convergence (Gaussian smoothing makes $q_\varepsilon$ smooth and positive), we obtain

$$\frac{dF}{d\varepsilon}\Big|_{\varepsilon=0} = \int \left( 1 + \log \frac{q_\rho}{\nu} \right) \delta q \, du = \int \left( 1 + \log \frac{q_\rho}{\nu} \right) (\xi * \varphi_H) \, du.$$

By Fubini/Tonelli and the adjointness of convolution,

$$\int (\xi * \varphi_H) \, g \, du = \int \xi \, (\varphi_H * g) \, du \qquad (\forall \, g),$$

hence

$$\frac{dF}{d\varepsilon}\Big|_{\varepsilon=0} = \int \left[ \varphi_H * (1 + \log(q_\rho/\nu)) \right](u) \, \xi(u) \, du.$$

By the definition of the Gateaux derivative, we have that $\frac{\delta}{\delta \rho} \mathrm{KL}(q_\rho \| \nu) = \varphi_H * \left( 1 + \log(q_\rho/\nu) \right)$.

**Step 3: Spatial gradient of the first variation.** Since $\varphi_H$ is $C^\infty$ and convolution commutes with differentiation, $\nabla(\varphi_H * f) = \varphi_H * \nabla f$. Applying this to $f = 1 + \log(q_\rho/\nu)$ and using $\nabla \log(q_\rho/\nu) = \nabla \log q_\rho - \nabla \log \nu$ yields the equation

$$\nabla \frac{\delta}{\delta \rho} \mathrm{KL}(q_\rho \| \nu) = \varphi_H * \left( \nabla \log q_\rho - \nabla \log \nu \right). \tag{10}$$

**Step 4: Entropy.** For $G(\varepsilon) = \mathrm{Ent}(\rho_\varepsilon) = \int (\rho + \varepsilon \xi) \log(\rho + \varepsilon \xi) \, du$,

$$\frac{dG}{d\varepsilon}\Big|_{\varepsilon=0} = \int (1 + \log \rho) \, \xi \, du,$$

so $\frac{\delta}{\delta \rho} \mathrm{Ent}(\rho) = \log \rho + 1$ and taking $\nabla$ gives the equation

$$\nabla \frac{\delta}{\delta \rho} \mathrm{Ent}(\rho) = \nabla \log \rho. \tag{11}$$

Combining equations (10) and (11), we obtain the conclusion. $\qquad \square$

## A.3 Velocity Space and $\Sigma$-weighted Wasserstein Inner Product

**Notation and standing assumptions.** Let $\mathcal{D}(\Omega) := C_c^\infty(\Omega)$ denote the space of smooth, compactly supported test functions on $\Omega$, and let $\mathcal{D}'(\Omega)$ be its dual (the space of distributions). For $T \in \mathcal{D}'(\Omega)$ and $\varphi \in \mathcal{D}(\Omega)$ we write $\langle T, \varphi \rangle$ for the duality pairing. Let $\rho$ denote the current candidate density of the clean representation $h(X)$. Assume $\rho \in \mathcal{P}_2(\Omega)$. All integrals are with respect to Lebesgue measure $du$; $a \cdot b$ denotes the Euclidean inner product in $\mathbb{R}^{d_u}$.

*Standing boundary assumptions.* For any velocity field $w : \Omega \to \mathbb{R}^{d_u}$ considered below, we consider the following integrations by parts used below.

$$\int_\Omega \varphi \, \nabla \cdot (\rho w) \, du = \int_\Omega \left( \nabla \cdot (\varphi \, \rho w) - \rho w \cdot \nabla \varphi \right) du \quad \text{(product rule: } \nabla \cdot (\varphi F) = \varphi \, \nabla \cdot F + F \cdot \nabla \varphi)$$

$$= -\int_\Omega \rho \, w \cdot \nabla \varphi \, du + \int_\Omega \nabla \cdot (\varphi \, \rho w) \, du$$

$$= -\int_\Omega \rho \, w \cdot \nabla \varphi \, du + \int_{\partial \Omega} (\varphi \rho w) \cdot n \, dS \quad \text{(Gauss–Green)}.$$

$$= -\int_\Omega \rho \, w \cdot \nabla \varphi \, du$$

We will use one of the following assumptions (each suffices):

(A) **Test function case:** $\varphi \in C_c^\infty(\Omega)$. Then $\varphi|_{\partial \Omega} = 0$ (support is compactly contained in $\Omega$), so the boundary integral vanishes—no boundary assumption on $w$ is needed.

(B) **No-flux case (standing assumption):** if $\Omega$ is bounded and $(\rho w) \cdot n = 0$ on $\partial \Omega$, then the boundary term is $0$.

(C) **Whole space:** if $\Omega = \mathbb{R}^{d_u}$, then either $\rho w$ has compact support or decays sufficiently fast so that the flux at infinity vanishes (via cutoff arguments).

In each case we conclude

$$\int_\Omega \varphi \, \nabla \cdot (\rho w) \, du = -\int_\Omega \rho \, w \cdot \nabla \varphi \, du.$$

**Velocity space, mass–flux divergence, and pre-tangent set.** Let $\Sigma : \Omega \to \mathbb{R}^{d_u \times d_u}$ be a smooth, symmetric, positive-definite matrix field (the mobility). Given a (probability) mass density $\rho \in \mathcal{P}_2(\Omega)$ and a mobility $\Sigma$, define a $\Sigma$-weighted velocity space

$$L^2_{\rho, \Sigma^{-1}}(\Omega; \mathbb{R}^{d_u}) := \left\{ w : \Omega \to \mathbb{R}^{d_u} \; : \; \int_\Omega w^\top \Sigma^{-1} w \, \rho \, du < \infty \right\}.$$

Define the mass-flux divergence operator

$$K_\rho : \; L^2_{\rho, \Sigma^{-1}}(\Omega; \mathbb{R}^{d_u}) \longrightarrow \mathcal{D}'(\Omega), \qquad K_\rho w := -\nabla \cdot (\rho \, w).$$

For $\varphi \in \mathcal{D}(\Omega) = C_c^\infty(\Omega)$ we set

$$\langle K_\rho w, \varphi \rangle := \langle -\nabla \cdot (\rho w), \varphi \rangle = \int_\Omega \rho \, w \cdot \nabla \varphi \, du.$$

The pre-tangent set at $\rho \in \mathcal{P}_2(\Omega)$ is

$$T_\rho \mathcal{P}_2(\Omega) := \left\{ K_\rho w \; : \; w \in L^2_{\rho, \Sigma^{-1}}(\Omega; \mathbb{R}^{d_u}) \right\} = \left\{ -\nabla \cdot (\rho w) \; : \; w \in L^2_{\rho, \Sigma^{-1}}(\Omega; \mathbb{R}^{d_u}) \right\},$$

whose form coincides with the continuity equation $\partial_t \rho_t + \nabla \cdot (\rho_t w_t) = 0$ (see Appendix B.1).

**Velocity with minimal weighted kinetic energy.** Pick a tangent vector $\sigma \in T_\rho \mathcal{P}_2(\Omega)$, i.e. $\sigma = K_\rho w = -\nabla \cdot (\rho w)$ for some $w \in L^2_{\rho, \Sigma^{-1}}(\Omega; \mathbb{R}^{d_u})$. Among all velocities realizing $\sigma$, we want to find the one of minimal weighted kinetic energy via

$$\min_{w \in L^2_{\rho, \Sigma^{-1}}(\Omega; \mathbb{R}^{d_u})} \int_\Omega w^\top \Sigma^{-1} w \, \rho \, du \quad \text{subject to} \quad -\nabla \cdot (\rho w) = \sigma.$$

Introduce the Lagrangian with multiplier $\phi \in \{\phi \in L^2(\Omega) \ : \ \nabla\phi \in L^2(\Omega; \mathbb{R}^{d_u})\}$:

$$\mathcal{L}(w, \phi) = \int_\Omega \left( \tfrac{1}{2}\, w^\top \Sigma^{-1} w\, \rho + \phi\left[\sigma + \nabla\cdot(\rho w)\right]\right) du.$$

**Lemma 2** (Gâteaux derivative along $\eta$). *Suppose standing boundary assumptions hold. Let $w, \eta \in L^2_{\rho,\Sigma^{-1}}(\Omega; \mathbb{R}^{d_u})$, $\phi \in \{\phi \in L^2(\Omega) \ : \ \nabla\phi \in L^2(\Omega; \mathbb{R}^{d_u})\}$. Then*

$$\left.\frac{d}{d\varepsilon}\right|_{\varepsilon=0} \mathcal{L}(w + \varepsilon\eta, \phi) = \int_\Omega \rho\,(\Sigma^{-1} w - \nabla\phi)\cdot\eta\, du.$$

*Proof.* Differentiate the quadratic term:

$$\left.\frac{\mathrm{d}}{\mathrm{d}\varepsilon}\right|_{\varepsilon=0} \frac{1}{2}\, \rho\,(w + \varepsilon\eta)^\top \Sigma^{-1}(w + \varepsilon\eta) = \rho\,\eta^\top \Sigma^{-1} w$$

For the constraint term,

$$\left.\frac{d}{d\varepsilon}\right|_{\varepsilon=0} \int_\Omega \phi\, \nabla\cdot\left(\rho(w + \varepsilon\eta)\right) du = \int_\Omega \phi\, \nabla\cdot(\rho\eta)\, du = -\int_\Omega \rho\, \nabla\phi\cdot\eta\, du$$

by standing boundary assumptions. Summing gives the identity. $\qquad\square$

Since $\eta$ is arbitrary, Lemma 2 yields the optimal velocity $w^* = \Sigma\,\nabla\phi$ on $\{\rho > 0\}$. Enforcing the interior constraint with $w^* = \Sigma\nabla\phi$ gives $-\nabla\cdot\left(\rho\,\Sigma\,\nabla\phi\right) = \sigma$ in $\Omega$. In the classical $W_2$ setting (Ambrosio et al., 2008; corresponding to $\Sigma = I$), the tangent space at $\rho$ is defined as the $L^2(\rho)$-closure of gradient fields; among all velocity fields that realize the same distributional derivative and solve the continuity equation, one selects the unique minimal-norm representative. Lemma 2 is precisely the $\Sigma$-weighted analogue of this principle: given $\sigma = -\nabla\cdot(\rho w)$, we minimize the weighted kinetic energy $\int w^\top \Sigma^{-1} w\, \rho\, du$ over all $w$ realizing $\sigma$. This provides a natural generalization of the inner product in $L^2$ onto the $\Sigma$-weighted one under our standing boundary assumptions.

**$\Sigma$-weighted Wasserstein inner product.** For any tangent vector $\sigma = -\nabla\cdot(\rho\,\omega) \in T_\rho\mathcal{P}_2(\Omega)$, we can define its $\Sigma$-weighted Wasserstein norm by choosing $\omega$ as the minimizer $w^* = \Sigma\nabla\phi$ of the weighted kinetic energy:

$$\|\sigma\|^2_{\rho,\Sigma} = \int_\Omega (w^*)^\top \Sigma^{-1} w^*\, \rho\, du = \int_\Omega \nabla\phi^\top \Sigma\, \nabla\phi\, \rho\, du.$$

Moreover, for tangent vectors $\sigma_i = -\nabla\cdot(\rho\,\Sigma\nabla\phi_i)(i = 1, 2)$, the $\Sigma$-weighted Wasserstein inner product can be set

$$\langle \sigma_1, \sigma_2\rangle_{\rho,\Sigma} := \int_\Omega \nabla\phi_1^\top \Sigma\, \nabla\phi_2\, \rho\, du.$$

**Mass compatibility.** For any tangent vector $\sigma \in T_\rho\mathcal{P}_2(\Omega)$,

$$\int_\Omega \sigma\, du = 0,$$

since for $\sigma = -\nabla\cdot(\rho w)$ the divergence theorem and the standing boundary assumptions give $\int_\Omega \sigma\, du = -\int_{\partial\Omega}(\rho w)\cdot n\, dS = 0$.

### A.4 $\Sigma$-Weighted Wasserstein Riemannian gradient.

**First variation (Gâteaux derivative).** We say that the energy function $\mathcal{E}$ is Gâteaux differentiable at $\rho$ if there exists a (measurable) function $\psi$ such that for every *scalar* direction $\zeta \in C_c^\infty(\Omega)$ with the mass constraint $\int_\Omega \zeta\, du = 0$,

$$D\mathcal{E}(\rho)[\zeta] := \left.\frac{d}{d\varepsilon}\right|_{\varepsilon=0} \mathcal{E}(\rho + \varepsilon\zeta) = \int_\Omega \psi(u)\, \zeta(u)\, du.$$

We write $\psi = \frac{\delta\mathcal{E}}{\delta\rho}(\rho)$ (equivalently, $\psi(u) = \left(\frac{\delta\mathcal{E}}{\delta\rho}(\rho)\right)(u)$) and call $\psi$ the first variation; it is defined up to an additive constant due to the mass constraint.

**$\Sigma$-weighted Wasserstein Riemannian gradient.** Let $\psi = \frac{\delta \mathcal{E}}{\delta \rho}(\rho)$ and $\sigma = -\nabla \cdot (\rho \, \Sigma \nabla \phi) \in T_\rho \mathcal{P}_2(\Omega)$. Then, by definition of the Gâteaux derivative and our boundary assumptions,

$$D\mathcal{E}(\rho)[\sigma] = \int_\Omega \psi \, \sigma \, du = -\int_\Omega \psi \, \nabla \cdot (\rho \, \Sigma \nabla \phi) \, du = \int_\Omega \rho \, \nabla \psi^\top \Sigma \, \nabla \phi \, du.$$

Assuming the metric Dirichlet energy $\int_\Omega \rho \, \nabla \psi^\top \Sigma \, \nabla \psi \, du < \infty$, by Cauchy–Schwarz with respect to $\langle \cdot, \cdot \rangle_{\rho, \Sigma}$,

$$\left| D\mathcal{E}(\rho)[\sigma] \right| \leq \left( \int_\Omega \rho \, \nabla \psi^\top \Sigma \, \nabla \psi \, du \right)^{1/2} \|\sigma\|_{\rho, \Sigma}.$$

Thus $D\mathcal{E}(\rho)[\cdot]$ is a bounded linear functional on $T_\rho \mathcal{P}_2(\Omega)$. By the Riesz representation theorem there exists a unique $\Sigma$-weighted Wasserstein Riemannian gradient $\mathrm{grad}_{W_{2,\Sigma}} \mathcal{E}(\rho) \in \overline{T_\rho \mathcal{P}_2(\Omega)}$ at $\rho \in \mathcal{P}_2(\Omega)$ such that

$$D\mathcal{E}(\rho)[\sigma] = \langle \mathrm{grad}_{W_{2,\Sigma}} \mathcal{E}(\rho), \sigma \rangle_{\rho, \Sigma}$$

for all $\sigma \in T_\rho \mathcal{P}_2(\Omega)$. Recall the $\Sigma$-weighted inner product at $\rho$:

$$\langle \sigma_1, \sigma_2 \rangle_{\rho, \Sigma} = \int_\Omega \nabla \phi_1^\top \Sigma \, \nabla \phi_2 \, \rho \, du, \qquad \sigma_i = -\nabla \cdot (\rho \, \Sigma \nabla \phi_i).$$

And necessarily

$$\mathrm{grad}_{W_{2,\Sigma}} \mathcal{E}(\rho) = -\nabla \cdot \left( \rho \, \Sigma \, \nabla \psi \right) = -\nabla \cdot \left( \rho \, \Sigma \, \nabla \frac{\delta \mathcal{E}}{\delta \rho}(\rho) \right).$$

**Energy dissipation via steepest descent.** From the definition of $\Sigma$-weighted Wasserstein Riemannian gradient, for any $\sigma \in T_\rho \mathcal{P}_2(\Omega)$ :

$$D\mathcal{E}(\rho)[\sigma] = \langle \mathrm{grad}_{W_{2,\Sigma}} \mathcal{E}(\rho), \, \sigma \rangle_{\rho, \Sigma} \geq -\left\| \mathrm{grad}_{W_{2,\Sigma}} \mathcal{E}(\rho) \right\|_{\rho, \Sigma} \|\sigma\|_{\rho, \Sigma},$$

with equality iff $\sigma = -\lambda \, \mathrm{grad}_{W_{2,\Sigma}} \mathcal{E}(\rho)$ for some $\lambda \geq 0$.

Let $t \mapsto \rho_t$ be an curve in $(\mathcal{P}_2(\Omega), W_{2,\Sigma})$ and $\psi_t = \delta \mathcal{E} / \delta \rho (\rho_t)$. By the chain rule in Banach spaces and by defining the $\Sigma$-weighted Wasserstein Riemannian gradient as the Riesz representation with respect to $\langle \cdot, \cdot \rangle_{W_{2,\Sigma}, \rho_t}$, we have

$$\frac{d}{dt} \mathcal{E}(\rho_t) = D\mathcal{E}(\rho_t)[\partial_t \rho_t] = \langle \mathrm{grad}_{W_{2,\Sigma}} \mathcal{E}(\rho_t), \, \partial_t \rho_t \rangle_{\rho_t, \Sigma}.$$

By Cauchy–Schwarz,

$$\frac{d}{dt} \mathcal{E}(\rho_t) \geq -\left\| \mathrm{grad}_{W_{2,\Sigma}} \mathcal{E}(\rho_t) \right\|_{\rho_t, \Sigma} \left\| \partial_t \rho_t \right\|_{\rho_t, \Sigma},$$

with equality iff $\partial_t \rho_t = -\lambda_t \, \mathrm{grad}_{W_{2,\Sigma}} \mathcal{E}(\rho_t)$ for some $\lambda_t \geq 0$.

Choosing $\lambda_t = 1$, we adopt the (negative) Wasserstein Riemannian gradient

$$\partial_t \rho_t = -\mathrm{grad}_{W_{2,\Sigma}} \mathcal{E}(\rho_t)$$

as the instantaneous rate of change of the density. Hence,

$$\partial_t \rho_t = \nabla \cdot \left( \rho_t \, \Sigma \, \nabla \psi_t \right), \qquad \psi_t := \frac{\delta \mathcal{E}}{\delta \rho}(\rho_t).$$

Furthermore,

$$\frac{d}{dt} \mathcal{E}(\rho_t) = -\left\| \mathrm{grad}_{W_{2,\Sigma}} \mathcal{E}(\rho_t) \right\|_{\rho_t, \Sigma}^2 = -\int_\Omega \rho_t \, \nabla \psi_t^\top \Sigma \nabla \psi_t \, du \leq 0,$$

which is the energy-dissipation identity for $\Sigma$-weighted Wasserstein gradient flows; see Otto (2001) for the unweighted case $\Sigma \equiv I$. Thus $\mathcal{E}(\rho_t)$ is nonincreasing along the flow, and it is strictly decreasing unless $\nabla \psi_t = 0$ $\rho_t$-a.e.

As discussed above, we have the following theorem showing that the $\Sigma$-weighted Wasserstein Riemannian gradient achieves steepest descent under standing boundary assumptions.

**Theorem 1** (Steepest descent and energy dissipation in $W_{2,\Sigma}$). *Let $\mathcal{E} : \mathcal{P}_2(\Omega) \to \mathbb{R}$ be Gâteaux differentiable at $\rho$ with first variation $\psi = \delta\mathcal{E}/\delta\rho$ satisfying $\int_\Omega \rho \nabla\psi^\top \Sigma\nabla\psi \, du < \infty$. Suppose that the standing assumptions hold, then the $\Sigma$-weighted Wasserstein Riemannian gradient*

$$\mathrm{grad}_{W_{2,\Sigma}}\mathcal{E}(\rho) \;=\; -\nabla\cdot\big(\rho\,\Sigma\nabla\psi\big) \in T_\rho\mathcal{P}_2(\Omega)$$

*simultaneously characterizes the steepest descent and the energy dissipation as follows: for every $\sigma \in T_\rho\mathcal{P}_2(\Omega)$,*

$$D\mathcal{E}(\rho)[\sigma] = \big\langle \mathrm{grad}_{W_{2,\Sigma}}\mathcal{E}(\rho),\, \sigma \big\rangle_{\rho,\Sigma} \;\geq\; -\big\|\mathrm{grad}_{W_{2,\Sigma}}\mathcal{E}(\rho)\big\|_{\rho,\Sigma}\,\|\sigma\|_{\rho,\Sigma},$$

*with equality iff $\sigma = -\alpha\,\mathrm{grad}_{W_{2,\Sigma}}\mathcal{E}(\rho)$; hence among all unit directions the maximal instantaneous decrease of $\mathcal{E}$ is realized by $-\mathrm{grad}_{W_{2,\Sigma}}\mathcal{E}(\rho)$. Moreover, if $\rho_t$ solves the (negative) $\Sigma$-weighted Wasserstein gradient flow*

$$\partial_t\rho_t = -\mathrm{grad}_{W_{2,\Sigma}}\mathcal{E}(\rho_t) = \nabla\cdot\big(\rho_t\,\Sigma\nabla\psi_t\big), \qquad \psi_t = \frac{\delta\mathcal{E}}{\delta\rho}(\rho_t),$$

*then along the flow the energy dissipates according to*

$$\frac{d}{dt}\,\mathcal{E}(\rho_t) = -\big\|\mathrm{grad}_{W_{2,\Sigma}}\mathcal{E}(\rho_t)\big\|_{\rho_t,\Sigma}^2 = -\int_\Omega \rho_t\,\nabla\psi_t^\top\Sigma\nabla\psi_t\,du \;\leq 0,$$

*with equality iff $\nabla\psi_t = 0$ for $\rho_t$-a.e. points. Here $\|z\|_\Sigma^2 := z^\top\Sigma z$ and $\langle\cdot,\cdot\rangle_{\rho,\Sigma}$ is the inner product induced by $\Sigma$ at point $\rho$.*

**Remark on Wasserstein steepest descent and energy dissipation.** Our use of a Wasserstein gradient-flow viewpoint is inspired by the classical theory: informally, for an energy functional $\mathcal{E}$ on probability laws, one may write the evolution as a (negative) $W_2$ gradient flow so that the associated energy is nonincreasing along the flow, and equivalent formulations can be employed depending on convenience. Precise statements and various equivalent derivations (e.g., via Riesz representation in the $W_2$ geometry, the continuity equation, and the minimizing-movement scheme) can be found in the standard references Ambrosio et al. (2008); Otto (2001); Jordan et al. (1998); Santambrogio (2015); Peyré & Cuturi (2019).

# B   DESIGN OF THE PARTICLE FLOW ALGORITHM

## B.1   KINEMATIC VELOCITY AND PARTICLE FLOW UPDATE

**Motivation and overview.** In this section we optimize an unknown probability density $\rho \in \mathcal{P}_2(\Omega)$ rather than a vector in an Euclidean space. Hence updates must preserve mass and nonnegativity ($\rho \geq 0$, $\int_\Omega \rho = 1$), which suggests a kinematic viewpoint instead of a naive Euclidean gradient step. We introduce a time-dependent velocity field $v_t : \Omega \to \mathbb{R}^{d_u}$ that transports the probability density by the pushforward rule $\rho_t = (\Phi_t)_\# \rho_0$, where the flow map $\Phi_t$ solves the characteristics $\dot{X}_t = v_t(X_t)$, which is equivalently described by the continuity equation. Choosing the velocity to follow the $\Sigma$-weighted Wasserstein gradient of the energy yields the negative gradient flow, along which the energy decreases monotonically. In practice, we employ a simple particle approximation of the characteristics (explicit-Euler style), so that the empirical measure tracks the continuum evolution.

**Flow map and continuity equation.** Let $v_t : \Omega \to \mathbb{R}^{d_u}$ be a time-dependent velocity field. Consider the ordinary differential equation (ODE)

$$\dot{X}_t = v_t(X_t), \qquad X_0 \sim \rho_0.$$

Under some regular conditions (Ambrosio et al., 2008), the ODE admits a unique maximal solution and defines a unique flow map $\Phi_t(x)$ satisfying

$$\frac{d}{dt}\Phi_t(x) = v_t\big(\Phi_t(x)\big), \qquad \Phi_0(x) = x,$$

and $X_t = \Phi_t(X_0)$. Denote $\rho_t$ as the probability density of $X_t$. Furthermore, we have that

$$\int_\Omega \varphi(u)\,\rho_t(u)\,du = \int_\Omega \varphi(\Phi_t(x))\,\rho_0(x)\,dx,$$

for every test function $\varphi \in C_c^\infty(\Omega)$. Differentiating in $t$ (chain rule) gives

$$\frac{d}{dt}\int_\Omega \varphi(u)\,\rho_t(u)\,du = \int_\Omega \nabla\varphi(\Phi_t(x))\cdot v_t(\Phi_t(x))\,\rho_0(x)\,dx.$$

The change of variables $u = \Phi_t(x)$ yields

$$\frac{d}{dt}\int_\Omega \varphi(u)\,\rho_t(u)\,du = \int_\Omega \nabla\varphi(u)\cdot v_t(u)\,\rho_t(u)\,du.$$

Applying the product rule $\nabla\cdot(\varphi\,\rho_t v_t) = \nabla\varphi\cdot(\rho_t v_t) + \varphi\,\nabla\cdot(\rho_t v_t)$ and Gauss–Green, and using the standing boundary assumptions, we obtain

$$\frac{d}{dt}\int_\Omega \varphi\,\rho_t = -\int_\Omega \varphi\,\nabla\cdot(\rho_t v_t)\,du.$$

Since $\varphi$ is arbitrary, in the sense of distributions

$$\partial_t\rho_t + \nabla\cdot(\rho_t v_t) = 0 \qquad \text{(continuity equation)}.$$

This is a standard result; see Figalli & Glaudo (2021).

**Kinematic velocity induced by the Wasserstein Riemannian gradient.** As discussed in the last section, adopting the (negative) $\Sigma$-weighted Wasserstein gradient flow as the instantaneous evolution gives

$$\partial_t\rho_t = -\operatorname{grad}_{W_{2,\Sigma}}\mathcal{E}(\rho_t) = \nabla\cdot\big(\rho_t\,\Sigma\,\nabla\psi_t\big), \qquad \psi_t := \frac{\delta\mathcal{E}}{\delta\rho}(\rho_t).$$

Consequently, in kinematic form the associated steepest-descent velocity field is

$$v_t(u) := -\Sigma(u)\,\nabla\psi_t(u), \qquad u \in \Omega,$$

and the continuity equation $\partial_t\rho_t + \nabla\cdot(\rho_t v_t) = 0$ is precisely the Eulerian form of the gradient flow. This velocity field induces the particle characteristics $\dot{X}_t = v_t(X_t)$, whose time discretization yields the explicit Euler particle flow update.

**Explicit Euler particle flow update.** Represent the state at step $k$ by particles $\{\xi_i^k\}_{i=1}^N$. Take the explicit Euler method (Hairer et al., 1993) with step size $\tau_k > 0$:

$$\xi_i^{k+1} = \xi_i^k + \tau_k\,v\big(\xi_i^k\big) = \xi_i^k - \tau_k\,\Sigma^{(k)}\big(\xi_i^k\big)\,\nabla\psi^{(k)}\big(\xi_i^k\big), \qquad i = 1,\dots,N.$$

by using the steepest-descent velocity $v = -\Sigma\nabla\psi$. Using Lemma 1, we have

$$\nabla\psi^{(k)}(u) = \big[\varphi_H * (\nabla\log q_{\rho^{(k)}} - \nabla\log\nu)\big](u) + \lambda\,\nabla\log\rho^{(k)}(u), \tag{12}$$

with $q_{\rho^{(k)}} = \rho^{(k)} * \varphi_H$. Here $\rho^{(k)}$ can $\nu$ can be estimated by nonparametric method. Furthermore, define the three scores

$$s_q^{(k)}(u) := \nabla\log q_{\rho^{(k)}}(u),\ s_\nu(u) := \nabla\log\nu(u),\ s_\rho^{(k)}(u) := \nabla\log\rho^{(k)}(u), \tag{13}$$

with the smoothed density $q_{\rho^{(k)}} := \rho^{(k)} * \varphi_H$. The first–variation identity yields

$$\nabla\psi^{(k)}(u) = \big[\varphi_H * (s_q^{(k)} - s_\nu)\big](u) + \lambda\,s_\rho^{(k)}(u). \tag{14}$$

The explicit Euler particle update (step size $\tau_k > 0$) is

$$\xi_i^{k+1} = \xi_i^k - \tau_k\,\Sigma^{(k)}\big(\xi_i^k\big)\Big(\big[\varphi_H * (s_q^{(k)} - s_\nu)\big](\xi_i^k) + \lambda\,s_\rho^{(k)}(\xi_i^k)\Big), \qquad i = 1,\dots,N. \tag{15}$$

Furthermore, let $\tilde{s}_q^{(k)} := \varphi_H * s_q^{(k)}, \tilde{s}_\nu := \varphi_H * s_\nu$. Then equation (14) becomes exactly

$$\nabla\psi^{(k)} = \tilde{s}_q^{(k)} - \tilde{s}_\nu + \lambda\,s_\rho^{(k)}, \tag{16}$$

and the Euler update is equivalently

$$\xi_i^{k+1} = \xi_i^k - \tau_k\,\Sigma^{(k)}(\xi_i^k)\,\big(\tilde{s}_q^{(k)}(u) - \tilde{s}_\nu(u) + \lambda\,s_\rho^{(k)}(u)\big)\Big|_{u=\xi_i^k}. \tag{17}$$

This is only a change of notation: equation (17) is identical to equation (15).

**Constructing the mobility from the clean embedding and the noise kernel.** Given a probability density $\rho^{(k)} \in \mathcal{P}_2(\Omega)$, we consider a smooth (possibly anisotropic) mobility matrix $\Sigma^{(k)}$ from the density itself by the (kernel) local second-moment tensor. Specifically, we compute the local covariance at location $u$ by

$$C^{(k)}(u; H) = \frac{\int \varphi_H(u - \tilde{u}) \, [\tilde{u} - \mu_{\rho^{(k)}}(u; H)][\tilde{u} - \mu_{\rho^{(k)}}(u; H)]^\top \rho^{(k)}(\tilde{u}) \, d\tilde{u}}{\int \varphi_H(u - \tilde{u}) \, \rho^{(k)}(\tilde{u}) \, d\tilde{u}}.$$

where the local mean $\mu_{\rho^{(k)}}(u; H) = \frac{\int \varphi_H(u-\tilde{u}) \, \tilde{u} \, \rho^{(k)}(\tilde{u}) \, d\tilde{u}}{\int \varphi_H(u-\tilde{u}) \, \rho^{(k)}(\tilde{u}) \, d\tilde{u}}$. A geometry-consistent practical choice is to instantiate the mobility in equation 17 as $\Sigma^{(k)}(u) \propto C^{(k)}(u; H)$. This yields a position-dependent metric aligned with the local structure of $\rho$ at the noise kernel bandwidth $H$.

**KDE plug-in for the particle-flow update.** For the three scores 13, they are estimated by the kernel density estimators (KDE). Specifically, we estimate $s_\nu$ with a KDE fitted to the current data mini-batch, while $s_\rho^{(k)}$ and $s_q^{(k)}$ are computed from the current particle cloud $\{\xi_i^{(k)}\}_{i=1}^N$ (they are not re-fit from the batch). Plugging these score estimates into the gradient-velocity field yields the explicit-Euler particle update. This KDE plug-in ensures that the first variation $\delta \mathcal{E}/\delta \rho$ and its spatial gradient are well defined and numerically stable. We deliberately avoid the atomic empirical measure $\rho^{\text{em}} = \frac{1}{N} \sum_{i=1}^N \delta(u - \xi_i)$, whose log-density is $+\infty$ on atoms and $-\infty$ elsewhere; consequently $\text{Ent}(\rho^{\text{em}}) = -\infty$, so the entropy regularizer (and its gradient) would be ill-posed. The KDE constructions and bandwidth choices are detailed in the next subsection.

## B.2 DENOISING ALGORITHM VIA PARTICLE FLOW UPDATE

**Gaussian kernels.** For a positive definite bandwidth/covariance $H \in \mathbb{R}^{d \times d}$ and a displacement $\delta \in \mathbb{R}^d$, define

$$\varphi_H(\delta) := (2\pi)^{-\frac{d}{2}} |H|^{-\frac{1}{2}} \exp\Big(-\tfrac{1}{2} \delta^\top H^{-1} \delta\Big), \qquad \|\delta\|_{H^{-1}}^2 := \delta^\top H^{-1} \delta.$$

We use three bandwidths for distinct roles:

$$H_v, H_\rho, H_q \text{ (inner KDEs)}, \qquad H_q \text{ (outer smoothing)}.$$

In practice these are often isotropic: $H_\bullet = h_\bullet^2 I$ where $\bullet$ represents $v, \rho, q$.

**Inner KDEs and score formulas.** Given centers $C = \{c_i\}_{i=1}^N$ and bandwidth $H \succ 0$, define the KDE and its score at $u \in \mathbb{R}^d$:

$$\widehat{p}(u; C, H) = \frac{1}{N} \sum_{i=1}^N \varphi_H(u - v_i), \tag{18}$$

$$s_{\text{KDE}}(u; C, H) = \nabla_u \log \widehat{p}(u; V, H) = \sum_{i=1}^N \omega_i(u; C, H) \, H^{-1}(c_i - u) \tag{19}$$

where the weights $\omega_i(u; C, H) = \frac{\exp\left(-\frac{1}{2}\|u-c_i\|_{H^{-1}}^2\right)}{\sum_{k=1}^N \exp\left(-\frac{1}{2}\|u-c_k\|_{H^{-1}}^2\right)}$.

**Empirical KDE approximations.** Given a training mini-batch $\{y_m\}_{m=1}^M$, the current observed embedding $U_{\text{batch}} = \{u_m\}_{m=1}^M = \{h_\theta(y_m)\}_{m=1}^M \subset \mathbb{R}^d$ where $h_\theta : \mathcal{Y} \to \mathbb{R}^d$ is an encoder. We maintain a particle cloud $\boldsymbol{\xi} = \{\xi_i\}_{i=1}^N \subset \mathbb{R}^d$ with $N$ particles, which are used to approximate a clean embedding density $\rho$ in the embedding space. In particular, this particle cloud $\boldsymbol{\xi}$ is not reinitialized across mini-batches or epochs (it is a moving, cross-batch state).

With three positive-definite bandwidth matrices $H_v, H_\rho, H_q$, we define

$$\widehat{\nu}(u) = \widehat{p}(u; U_{\text{batch}}, H_v), \tag{20}$$

$$\widehat{\rho}(u) = \widehat{p}(u; \xi, H_\rho), \tag{21}$$

$$\widehat{q}_\rho(u) = (\varphi_{H_q} * \widehat{\rho})(u) = \frac{1}{N} \sum_{i=1}^N (\varphi_{H_q} * \varphi_{H_\rho})(u - \xi_i). \tag{22}$$

For Gaussian kernels, $\varphi_{H_q} * \varphi_{H_\rho} = \varphi_{H_q + H_\rho}$ (+ denotes matrix addition of bandwidth/covariance), hence the above equation (22) reduces to $\widehat{p}(u; \xi, H_{\text{eff}})$ with the effective bandwidth $H_{\text{eff}} = H_q + H_\rho$.

**Associated three scores.**

$$s_\nu(u) = \nabla_u \log \widehat{\nu}(u), \qquad s_\rho(u) = \nabla_u \log \widehat{\rho}(u), \qquad s_q(u) = \nabla_u \log \widehat{q}_\rho(u), \qquad (23)$$

which all follow the generic form in equation (19).

**Outer smoothing of score fields (convolution-based).** Let $F : \mathbb{R}^d \to \mathbb{R}^d$ be a discrete vector field tabulated at centers $C = \{c_i\}_{i=1}^K$ with values $F(c_i) \in \mathbb{R}^d$. With a positive-definite bandwidth $H_u$, define the convolution smoother

$$(\varphi_{H_q} * F)(u) := \frac{1}{K} \sum_{i=1}^K \varphi_{H_q}(u - c_i) \, F(c_i).$$

Using particle cloud $\boldsymbol{\xi} = \{\xi_i\}_{i=1}^N$ and current batch $U_{\text{batch}} = \{u_m\}_{m=1}^M$ as centers, we set:

$$\begin{aligned}
\tilde{s}_q(u) &:= (\varphi_{H_q} * s_q)(u) \quad \text{with centers} \quad \{\xi_i\}_{i=1}^N, \\
\tilde{s}_\nu(u) &:= (\varphi_{H_q} * s_\nu)(u) \quad \text{with centers} \quad \{u_m\}_{m=1}^M.
\end{aligned} \qquad (24)$$

**A–step (particle flow; no backprop).** Let $\Sigma : \mathbb{R}^d \to \mathbb{R}^{d \times d}$ be the mobility and $\eta > 0$ a stepsize. Perform $S$ explicit Euler steps:

$$\boldsymbol{\xi} \leftarrow \boldsymbol{\xi} - \eta \left[ \Sigma(\boldsymbol{\xi}) \big( \tilde{s}_q(\boldsymbol{\xi}) - \tilde{s}_\nu(\boldsymbol{\xi}) + \lambda \, s_\rho(\boldsymbol{\xi}) \big) \right] \quad (25)$$

where the mobility $\Sigma(\boldsymbol{\xi})$ and three scores are evaluated pointwise at particle locations $\{\xi_i\}$. This update is computed with gradients disabled and with $U_{\text{batch}}$ treated as constants; hence it does not update $\theta$ in the decoder. The A–step performs $S$ explicit Euler updates per outer iteration (one batch). In practice we do not drive the particle flow to convergence inside a batch; instead we use a small fixed $S$ and rely on alternating updates across batches to gradually transport the particles.

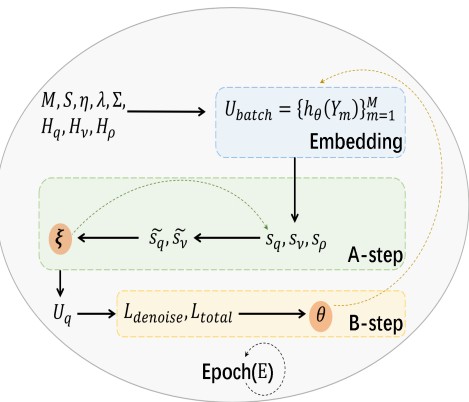

Figure 6: Illustration of the algorithm.

**B–step (KDE pullback to the student; backprop enabled).** The set

$$U_q = \left\{ \xi_i + \zeta_i : \zeta_i \sim \mathcal{N}(0, H_q) \right\}_{i=1}^N \qquad (26)$$

consists of i.i.d. samples from the empirical convolved density $\widehat{q}_\rho(u) = (\varphi_{H_q} * \widehat{\rho})(u) = \frac{1}{N} \sum_{i=1}^N (\varphi_{H_q} * \varphi_{H_\rho})(u - \xi_i)$. When the particle cloud $\boldsymbol{\xi}$ well approximates the target $\rho$, one has $\widehat{q}_\rho \approx q_\rho := \varphi_{H_q} * \rho$, and thus $U_q$ can be regarded as approximate samples from $q_\rho$. Consequently, the integrals in the denoising objective function (7) can be evaluated by Monte–Carlo averages over $U_q$ (and by a leave–one–out average over $\boldsymbol{\xi}$ for the entropy term), which converge to their population values by the law of large numbers.

Define a batch denoising objective $\text{KL}(q_\rho \| \nu) + \lambda \, \text{Ent}(\rho)$ by computing the KDEs from the latest $\boldsymbol{\xi}$:

$$L_{\text{denoise}} = \frac{1}{|U_q|} \sum_{u \in U_q} \left[ \log \widehat{q}_\rho(u) - \log \widehat{\nu}(u) \right] + \lambda \, \widehat{\text{Ent}}_{\text{LOO}}(\boldsymbol{\xi}; H_\rho). \qquad (27)$$

where the leave-one-out entropy is

$$\widehat{\text{Ent}}_{\text{LOO}}(\boldsymbol{\xi}; H_\rho) := -\frac{1}{N} \sum_{i=1}^N \log \widehat{\rho}^{\text{LOO}}(\xi_i; \boldsymbol{\xi} \setminus \{\xi_i\}, H_\rho),$$

---

**Algorithm 1:** Particle-flow denoising (outer mini-batch loop with $S$ inner Euler steps)

---

**Input:** epochs $E$, mini-batch size $M$, inner steps $S$, step size $\eta$, bandwidths $H_q, H_v, H_\rho$,
       regularizer $\lambda$, mobility $\Sigma$

**Output:** particle cloud $\xi$, encoder parameters $\theta$

1 **for** *epoch* $= 1$ **to** $E$ **do**

2     **foreach** *mini-batch* $\mathcal{B} = \{Y_m\}_{m=1}^M$ **do**

        `// (1) Embeddings`

3         $U_{\text{batch}} \leftarrow \{h_\theta(Y_m)\}_{m=1}^M$;

        `// (2) A-step (no grad):   ` $s = 1 \dots S$

4         **for** $s = 1$ **to** $S$ **do**

5             build $s_q, s_\nu, s_\rho$ by equation (23);

6             $\widetilde{s}_q \leftarrow \varphi_{H_q} * s_q, \;\; \widetilde{s}_\nu \leftarrow \varphi_{H_q} * s_\nu$ by equation (24);

7             $\boldsymbol{\xi} \leftarrow \boldsymbol{\xi} - \eta \, \Sigma(\boldsymbol{\xi})\big(\widetilde{s}_q(\boldsymbol{\xi}) - \widetilde{s}_\nu(\boldsymbol{\xi}) + \lambda s_\rho(\boldsymbol{\xi})\big)$ ;         `// equation (25)`

        `// (3) B-step (with grad)`

8         sample $U_q$ by equation (26);

9         form $L_{\text{denoise}}$ by equation (27); set $L_{\text{total}}$ by equation (28); backprop to update $\theta$;

---

with

$$\widehat{\rho}^{\text{LOO}}\big(\xi_i; \boldsymbol{\xi} \setminus \{\xi_i\}, H_\rho\big) := \frac{1}{N-1} \sum_{\substack{m=1 \\ m \neq i}}^N \varphi_{H_\rho}\big(\xi_i - \xi_m\big).$$

*Backprop paths.* Only the term $\log \widehat{\nu}(u) = \log \widehat{p}(\cdot; U_{\text{batch}}, H_v)$ backpropagates to $\theta$ via the centers $U_{\text{batch}} = \{h_\theta(y_m)\}_{m=1}^M$. Terms (those depending on $\boldsymbol{\xi}$) including $\log \widehat{q}_\rho(u)$ and $\widehat{\text{Ent}}_{\text{LOO}}(\boldsymbol{\xi}; H_\rho)$ do not backpropagate to $\theta$ (stop-gradient).

**Total loss and parameter update.** Let $L_{\text{main}}(\theta)$ be the main task objective function. We optimize

$$L_{\text{total}}(\theta) = L_{\text{main}}(\theta) + \tau_3 \, L_{\text{denoise}}(\theta), \tag{28}$$

and update $\theta$ by backpropagation through equation (28). Particles $\boldsymbol{\xi}$ are updated only in the A–step equation (25).

**Training loop (per iteration).**

1. **Embeddings.** Compute $U_{\text{batch}} = \{h_\theta(Y_m)\}_{m=1}^M$.

2. **A–step (no grad).** Build $s_q, s_\nu, s_\rho$ via equation 23; smooth to $\widetilde{s}_q, \widetilde{s}_\nu$ via equation 24; update the particles $\boldsymbol{\xi}$ by equation (25).

3. **B–step (with grad).** sample $U_q$ by equation (26); Form $L_{\text{denoise}}$ in equation (27) using $U_q$ and $U_{\text{batch}}$; set $L_{\text{total}}$ by equation (28); backpropagate to update $\theta$.

**Isotropic special case and its adaptive bandwidth.** Assume $H_q = h_q^2 I$, $H_v = h_v^2 I$, $H_\rho = h_\rho^2 I$. Here $h_\rho^2$ controls the intrinsic smoothness of the clean particle density $\rho$, $h_q^2$ controls the additional convolutional blur that turns $\rho$ into the observable model $q_\rho$; together they form the effective bandwidth $h_{\text{eff}}^2 = h_q^2 + h_\rho^2$. And $h_\nu^2$ stabilizes the KDE approximation of the batch density $\nu$. For a set of centers $C = \{c_i\}_{i=1}^K$, define the isotropic Gaussian weights

$$\omega_i^{(h)}(u) = \frac{\exp\big(-\|u - c_i\|^2/(2h^2)\big)}{\sum_{j=1}^K \exp\big(-\|u - c_j\|^2/(2h^2)\big)}.$$

The KDE score with bandwidth $h$ simplifies to

$$s_{\text{KDE}}^{(h)}(u; C) = \frac{\sum_i \omega_i^{(h)}(u)\, c_i - u}{h^2}.$$

With particle centers $\xi = \{\xi_i\}_{i=1}^N$ and the current batch $U_{\text{batch}} = \{u_m\}_{m=1}^M$,

$$\widehat{s}_q(u) = s_{\text{KDE}}^{(h_q)}(u; \xi), \qquad \widehat{s}_\nu(u) = s_{\text{KDE}}^{(h_v)}(u; U_{\text{batch}}), \qquad \widehat{s}_\rho(u) = s_{\text{KDE}}^{(h_\rho)}(u; \xi).$$

Let $w_i(u) = \exp\left(-\|u - c_i\|^2/(2h_q^2)\right)$ be the unnormalised kernel weights for the outer bandwidth. The outer convolutional smoother is

$$(\varphi_{h_q^2 I} * F)(u) = \frac{1}{K} \sum_{i=1}^K w_i(u)\, F(c_i).$$

We apply it to the score fields:

$$\tilde{s}_q(u) = (\varphi_{h_q^2 I} * \widehat{s}_q)(u) \ \text{ with } c_i = \xi_i, \qquad \tilde{s}_\nu(u) = (\varphi_{h_q^2 I} * \widehat{s}_\nu)(u) \ \text{ with } c_i = u_i.$$

With step size $\eta > 0$, the explicit Euler particle update reads

$$\xi_i^{k+1} = \xi_i^k - \eta \Sigma^{(k)}(\xi_i^k)\left[\tilde{s}_q(\xi_i^k) - \tilde{s}_\nu(\xi_i^k) + \lambda \widehat{s}_\rho(\xi_i^k)\right], \quad i = 1, \ldots, N,$$

where the mobility $\Sigma^{(k)}(u) \propto C^{(k)}(u; h_q^2 I)$. Furthermore, samples from $q_\rho = \rho * \varphi_{h_q^2 I}$ are approximated by

$$U_q = \left\{ \xi_i + \zeta_i \ : \ \zeta_i \sim \mathcal{N}(0, h_q^2 I) \right\}_{i=1}^N.$$

We adapt the three scalar bandwidths $h_\rho^2,\ h_q^2,\ h_\nu^2$ by the following objectives:

(A) **Clean-density bandwidth $h_\rho^2$ (for $\rho$).** Choose $h_\rho^2$ by maximizing the leave-one-out (LOO) log-likelihood of the particle cloud:

$$\widehat{h}_\rho^2 \in \arg\max_{h>0} \ \frac{1}{N} \sum_{i=1}^N \log\left[\frac{1}{N-1} \sum_{j \neq i} \varphi_{h^2}(\xi_i - \xi_j)\right].$$

This selects the kernel width that best fits the empirical $\rho$.

(B) **Convolution bandwidth $h_q^2$ (for $q_\rho$).** Given $\widehat{h}_\rho^2$, choose $h_q^2$ to minimize the negative log-likelihood of the observational batch under the convolved model $\widehat{q}_\rho$:

$$\widehat{h}_q^2 \in \arg\min_{h>0} -\frac{1}{M} \sum_{m=1}^M \log\left[\frac{1}{N} \sum_{i=1}^N \varphi_{h^2 + \widehat{h}_\rho^2}(u_m - \xi_i)\right].$$

This directly targets the cross-entropy $\mathbb{E}_{u \sim \nu}\left[-\log \widehat{q}_\rho(u)\right]$, i.e., the part of $\text{KL}(\nu \| \widehat{q}_\rho)$ that depends on $\widehat{q}_\rho$. It matches the convolved particle model to the observed embeddings while keeping $\rho$'s smoothing fixed.

(C) **Observation bandwidth $h_\nu^2$ (for $\nu$).** Independently tune $h_\nu^2$ using the LOO criterion on $U_{\text{batch}}$:

$$\widehat{h}_\nu^2 \in \arg\max_{h>0} \ \frac{1}{M} \sum_{m=1}^M \log\left[\frac{1}{M-1} \sum_{k \neq m} \varphi_{h^2}(u_m - u_k)\right].$$

This yields a stable self-density estimate $\widehat{\nu}(\cdot) = \widehat{p}(\cdot; U_{\text{batch}}, \widehat{h}_\nu^2)$ used in the KL term.

# C  ADDITIONAL EXPERIMENTAL RESULTS

## C.1  IMPLEMENTATION DETAILS

**Hardware and Environment.** Computational Setup: All experiments were performed using NVIDIA® RTX A6000 (48 GB) and GeForce RTX 5090 (32 GB) GPUs.

**License and open source notice.** Code released under the MIT License; dependent open-source libraries (e.g., PyTorch, causalchamber) follow their respective licenses. Data processing and model generation reference open-source implementations, with specific citations provided in related literature.

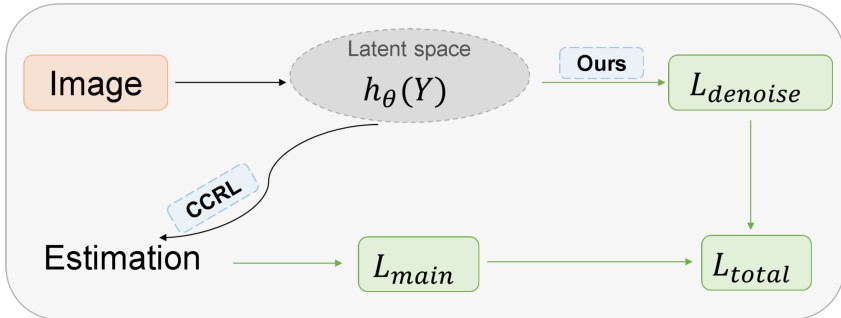

Figure 7: Overall framework of our method. The input image is mapped into a latent space representation, where we introduce the denoising loss $L_{\text{denoise}}$. This loss and the CCRL objective function $L_{\text{main}}$ are combined into the total loss $L_{\text{total}}$ for training.

## C.2 DATASETS

The dataset lt_camera_v1 comprises meticulously designed subsets: scm_2, scm_4, and scm_5. These subsets provide a multidimensional testing environment for algorithm evaluation by systematically varying the complexity of the causal graph, the number of variables, and the type of noise distribution.

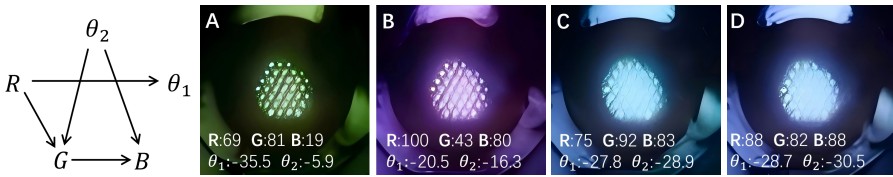

Figure 8: The ground-truth graph for scm_2. (A–D) Real images acquired from the scm_2 dataset, illustrating the influence of both the light source color $(R, G, B)$ and the linear polarizer angles $(\theta_1, \theta_2)$.

The DAG of scm_2 subset over $Z = (R, G, B, \theta_1, \theta_2)^\top$, causal relations are encoded by the graph shown in the left panel of Figure 8. The identical affine mapping to physical ranges and the same sampling/row normalization for $W$ and $D$—but scales each experiment to $N = 10,000$ samples (for 60,000 in total; see panels A–D of Figure 8). The sole substantive change is the exogenous noise: in the observational run, each component is drawn from a truncated normal on $[0, 1]$ with variable-specific means and standard deviations specified in the configuration, replacing the Uniform $[0, 1]$ noise. Interventions remain perfect—when targeting index $i$, all incoming edges to the target are removed and the mean of its noise term is shifted to a preset value (others unchanged). This shift to truncated Gaussian noise produces more sensor-like variability and makes scm_2 a stricter stress test for algorithms sensitive to noise assumptions (e.g. those relying on non-Gaussianity).

In contrast to the five-variable graphs used in scm_2, the scm_4 subset simplifies the setting by fixing the two polarizer variables $(\theta_1, \theta_2)$ to zero. The latent DAG thus reduces to three nodes with a causal chain $R \to G \to B$. All other settings follow scm_2 in brief: fixed camera parameters, the same affine mapping to physical ranges, and the same sampling/row-normalization of $W$ and $D$. Exogenous noises are truncated normal on $[0, 1]$ with variable-specific means and standard deviations specified in the configuration; interventions are perfect—when targeting $R$, $G$, or $B$, all incoming edges to the target are removed and the mean of its noise term is shifted to a preset value (others unchanged). The subset comprises one observational and three single-node interventional datasets, each with $N = 10,000$ samples, totaling 40,000 (see panels A–C of Figure 9).

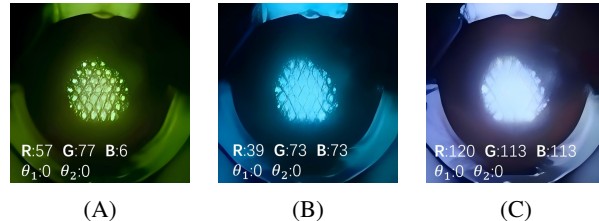

Figure 9: Real images acquired from the scm_4 dataset. The linear polarizer angles $(\theta_1, \theta_2)$ are fixed at $0°$, which highlights the influence of the light source color $(R, G, B)$.

Similar to scm_4, the scm_5 subset fixes the polarizers at $\theta_1 = \theta_2 = 0$ and otherwise retains the same settings (camera configuration, an affine mapping to physical ranges, and the same sampling/row-normalization scheme for $W$ and $D$). The latent variables reduce to $Z = (R, G, B)^\top$, but the underlying DAG is a collider $R \to G \leftarrow B$. In the observational regime, exogenous noises are independent truncated normals on $[0, 1]$ with variable-specific means and standard deviations specified in the configuration; accordingly, $R$ and $B$ are marginally independent yet become dependent when conditioning on $G$ (explaining away), offering a targeted test of v-structure recovery. Interventions are perfect: when targeting $i \in \{R, G, B\}$, all incoming edges to $i$ are removed and the mean of its noise term is shifted to a preset value (others unchanged), after which the same affine mapping is applied. The subset comprises one observational and three single-node interventional datasets, each with $N = 10{,}000$ samples, totaling $40{,}000$ (see panels A–C of Figure 10).

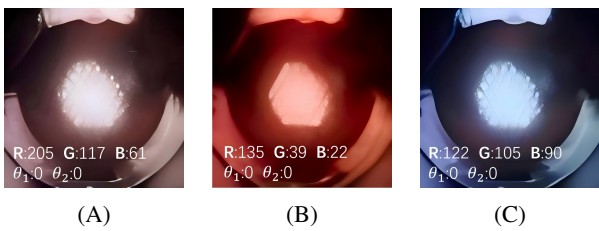

Figure 10: Real images acquired from the scm_5 dataset. The linear polarizer angles $(\theta_1, \theta_2)$ are fixed at $0°$, which highlights the influence of the light source color $(R, G, B)$.

### C.3 EVALUATION METRICS

**Mean Correlation Coefficient (MCC).** MCC (Hyvarinen & Morioka, 2016; Khemakhem et al., 2020) measures the pairwise correlation between the learned representation and the ground truth latents. Let $C \in \mathbb{R}^{d \times d}$ be the Pearson correlation matrix between the learned embeddings and the ground-truth variables. $\mathfrak{S}_d$ denotes the set of $d$-permutations. The MCC score is formally defined as

$$\text{MCC} = \max_{\pi \in \mathfrak{S}_d} \frac{1}{d} \sum_{j=1}^{d} \left| C_{j, \pi(j)} \right|,$$

where the maximum is taken over all permutations $\pi$ of $d$ elements.

**Structural Hamming Distance (SHD).** The SHD (Tsamardinos et al., 2006) measures how well a method recovers the ground-truth causal graph by comparing the adjacency matrix of the true graph $A \in \{0, 1\}^{d \times d}$ with that of the estimated graph $\hat{A} \in \{0, 1\}^{d \times d}$. It is defined as the number of edge insertions, deletions, or orientation errors required to transform $\hat{A}$ into $A$. Formally,

$$\text{SHD} := \sum_{i,j} \left| A_{i,j} - \hat{A}_{i,j} \right|.$$

By convention, a nonzero entry $A_{i,j} \neq 0$ indicates a directed edge $i \to j$. Since the Contrastive CRL method outputs a continuous estimate of the adjacency matrix, thresholding is required to obtain binary graphs. Following common practice, the threshold is chosen to match the number of edges in the ground-truth graph whenever known. A smaller SHD indicates a closer match between the estimated and true causal structures.

**Area Under the Receiver Operating Characteristic Curve (AUROC).** AUROC provides a comprehensive measure of the trade-off between the True Positive Rate (TPR) and the False Positive Rate (FPR) across different thresholds (Hanley & McNeil, 1982). TPR and FPR are defined as follows:

$$TPR = \frac{TP}{TP + FN}; FPR = \frac{FP}{FP + TN}$$

where TP denotes the number of true positive edges, FP the number of false positive edges, FN the number of false negative edges, and TN the number of true negative edges. To compute AUROC, we evaluate the off-diagonal entries of the estimated adjacency matrix, calculate TPR and FPR at different thresholds, plot the ROC curve, and integrate the area under it. Formally,

$$AUROC = \int_0^1 TPR(FPR)\, d FPR$$

An AUROC value of 0.5 indicates random prediction, while values closer to 1.0 indicate higher predictive accuracy.

**The Pearson Correlation Coefficient (PCC).** PCC is a statistical index used to measure the strength and direction of the linear relationship between two variables. The calculation formula is as follows:

$$PCC = \frac{\sum_{i=1}^{N}(x_i - \bar{x})(y_i - \bar{y})}{\sqrt{\sum_{i=1}^{N}(x_i - \bar{x})^2}\sqrt{\sum_{i=1}^{N}(y_i - \bar{y})^2}}$$

where N is the total number of samples, $x_i$ is the predicted value, $y_i$ is the true value, and $\bar{x}$ and $\bar{y}$ are their respective means. The value of PCC ranges from -1 to +1, where +1, -1, and 0 represent a perfect positive correlation, a perfect negative correlation, and no linear relationship, respectively.

**Mean Squared Error (MSE).** MSE is a common metric to measure the difference between predicted and true values. Its purpose is to quantify the magnitude of the error, and it calculates the average of the squared prediction errors. MSE is defined as

$$MSE = \frac{1}{N}\sum_{i=1}^{N}(x_i - y_i)^2$$

where N is the total number of samples, $x_i$ is the predicted value, and $y_i$ is the true value. MSE calculates the average of the squared prediction errors, thus placing a higher weight on larger errors.

**Normalized Mutual Information (NMI).** NMI is a metric used to measure the similarity between two clustering results (Strehl & Ghosh, 2002) and is generally defined as

$$NMI(X, Y) = \frac{2 \cdot I(X;Y)}{H(X) + H(Y)}$$

where $X$ and $Y$ represent the two clustering results, respectively. $I(X;Y)$ is the mutual information, which represents the reduction in uncertainty about one clustering result given the knowledge of the other. $H(X)$ and $H(Y)$ are the entropies of the two clustering results, measuring their respective uncertainties. The value of NMI ranges from 0 to 1, where 1 signifies that the two clustering results are identical, and 0 means that there is no shared information between them.

**The Adjusted Rand Index (ARI).** The ARI is an improved version of the Rand Index (RI)(Hubert & Arabie, 1985). The RI measures similarity by calculating the number of pairs of data points that are treated consistently in both clusterings (that is, belonging to the same cluster or to different clusters). However, the RI does not correct for the expected agreement that might occur by chance. The ARI addresses this issue by introducing a correction for chance, and its general form is the following.

$$ARI = \frac{\text{Index} - \text{Expected Index}}{\text{Max Index} - \text{Expected Index}}$$

where "Index" is the original RI and "Expected Index" is the expected value of the RI under the assumption that the two clusterings are random partitions. The value of the ARI typically ranges from -1 to 1.

**Marker score.** The Marker gene score is a proxy metric for the fine-grained preservation of biological signals (Gao et al., 2025), calculated with the formula:

$$\text{Marker gene score} = \frac{1}{K} \sum_{i=1}^{K} \frac{\left| G_x^i \cap G_y^i \right|}{\left| G_x^i \cup G_y^i \right|}$$

where K is the total number of classes in the data; $G_x^i$ is the set of marker genes identified for the $i$-th cell subpopulation of the true gene expression matrix; $G_y^i$ is the set of marker genes identified for the same $i$-th cell subpopulation from the predicted gene expression matrix; $G_x^i \cap G_y^i$ is the number of marker genes commonly identified in both the true and generated data for the $i$-th cell type and $G_x^i \cup G_y^i$ is the total number of unique genes identified as marker genes in either the true or the generated data for the $i$-th cell type.

## C.4 Baseline Details

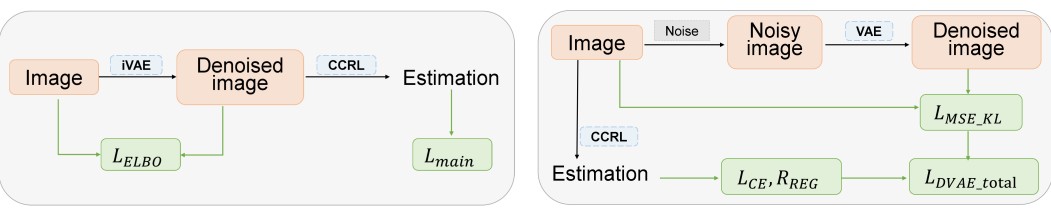

(a) iCCRL Framework          (b) DCCRL Framework

Figure 11: Overall frameworks of two methods. (a) iCCRL first performs latent-space denoising by optimizing the conditional ELBO ($L_{ELBO}$) and then feeds the denoised images into CCRL for causal-structure estimation ($L_{main}$). (b) DCCRL synthesizes noisy inputs, trains a VAE with an MSE+KL reconstruction objective ($L_{MSE\_KL}$), and runs CCRL with cross-entropy and regularization ($L_{CE}$, $R_{REG}$); the overall DVAE objective is $L_{DVAE\_total}$. Green boxes denote loss terms and arrows indicate data flow.

**Contrastive Causal Representation Learning (CCRL).** CCRL (Buchholz et al., 2023; Gamella et al., 2025a) assumes latent variables follow a linear-Gaussian SCM and observations are generated by a general nonlinear mixing. It leverages multi-environment data with single-node interventions on the latents. The method casts observational-interventional discrimination as a contrastive classification problem and uses a quadratic readout to approximate the Bayesian log-likelihood ratio. With sufficient intervention coverage and model capacity, CCRL achieves identifiability under general nonlinear mixing—up to permutation and scaling—without requiring paired counterfactuals.

**iVAE CCRL (iCCRL).** The Identifiable Variational Autoencoder (iVAE) (Khemakhem et al., 2020) extends the standard VAE (Kingma & Welling, 2019) by introducing a prior determined by an auxiliary variable $u$. For noisy data, the model assumes that observations are generated from latent signals transformed by a nonlinear mapping with added independent noise. During inference, denoising is achieved by maximizing a conditional ELBO.

**DVAE CCRL (DCCRL).** The Denoising Variational Autoencoder (DVAE) (Im et al., 2017) is a generative model that integrates the principles of both Denoising Autoencoders (DAE) (Vincent et al., 2008) and Variational Autoencoders (VAE) (Kingma & Welling, 2019). By encoding noisy input data and learning to reconstruct the clean original data, this approach compels the model to learn more robust and essential latent representations of the underlying data distribution.

**CausCell.** CausCell (Gao et al., 2025) is a single-cell representation learning framework that integrates structural causal models with diffusion models to achieve causal disentanglement and controllable counterfactual generation. By constructing causal graphs among latent biological concepts and guiding diffusion-based denoising, it generates gene expression profiles consistent with causal constraints while preserving fine-grained cellular variations. Modeling both observed and unexplained concepts, CausCell improves the interpretability, generalizability, and controllability of single-cell representations.

## C.5 HYPER-PARAMETERS

- The hyperparameter configuration for CCRL is as follows: 100 training epochs; batch size selected from 64, 512; and a latent dimensionality of 5.

- To control the MLP complexity in the iVAE component while maintaining training efficacy, we adopt the following hyperparameters: MLP depth of 5 hidden layers; hidden dimensionality chosen from 64, 128, 512 ; number of training epochs chosen from 50, 80, 100, 120; and a fixed batch size of 64. For the CCRL module, we train for 100 epochs, with the batch size selected from 64, 128, 512.

- DVAE-CCRL mirrors the CCRL baseline: training runs for 100 epochs with mini-batches of 64 or 512, a learning rate of $5 \times 10^{-4}$, and a 5-dimensional latent (hidden) layer.

- To balance denoising strength and training stability in the latent-space particle denoising module, while preserving identifiability in structural learning, we adopt the following hyperparameter settings. In the free-energy objective, the weight of the denoising term $\tau_3$ ranges from 0.1 to 1, and the entropy regularization coefficient $\lambda$ ranges from 0.001 to 0.5. In each iteration, the number of inner Euler steps for particle updates is chosen from 1, 2, 3, and the Euler step size ranges from 0.01 to 1. The learning rate of the structural (parametric) branch is set in the range $1 \times 10^{-4}$ to 0.1. For kernel density estimation (KDE), the number of particles per environment (including both observational and interventional settings) is set between 512 and 896, with 512 as the default.

- For the MERFISH Brain Dataset, all hyperparameters remain consistent with the CausCell baseline. Training is conducted for 5000 steps with gradient accumulation every 2 steps. For our proposed method, we adopt the following hyperparameter configuration: the denoising free-energy weight is set to $\tau_3 = 0.3$ and the entropy regularization coefficient to $\lambda = 0.003$. The inner loop performs a single explicit Euler update with a step size of 0.1, and the number of particles is fixed at 64.

## C.6 MORE RESULTS

### C.6.1 LT_CAMERA_V1 DATASET

We evaluated four methods in the datasets scm_2, scm_4, scm_5, which differ from lt_crl_benchmark_v1 by replacing uniform noise with truncated Gaussian noise, posing a stricter challenge for noise-dependent algorithms. The results are shown in Table 6, Table 7, Table 8.

| Metric | CCRL | iCCRL | DCCRL | Ours |
|---|---|---|---|---|
| MCC↑ | $0.0124 \pm$ 0.0026 | $0.0198 \pm$ 0.0062 | $0.0198 \pm$ 0.0066 | $\mathbf{0.0232} \pm$ **0.0062** |
| SHD↓ | $2.8000 \pm$ 0.4472 | $2.6000 \pm$ 0.8944 | $\mathbf{1.8000} \pm$ **0.4472** | $2.8000 \pm$ 0.4472 |
| AUROC↑ | $0.7353 \pm$ 0.0171 | $\mathbf{0.8413} \pm$ **0.0491** | $0.7933 \pm$ 0.0189 | $0.8227 \pm$ 0.0413 |

Table 6: Comparison of methods across metrics on Real-World Dataset scm_2.

Table 6 presents the results on the real dataset scm_2, where the truncated Gaussian noise exposes clear performance differences among methods. Our approach achieves the highest MCC, DCCRL attains the best SHD, and iCCRL records the highest AUROC. In contrast, CCRL performs worst across all metrics, highlighting its limited robustness under complex noise conditions.

| Metric | CCRL | iCCRL | DCCRL | Ours |
|---|---|---|---|---|
| MCC↑ | $0.0233 \pm$ 0.0100 | $0.0251 \pm$ 0.0037 | $0.0221 \pm$ 0.0103 | $\mathbf{0.0267} \pm$ **0.0062** |
| SHD↓ | $1.0000 \pm$ 0.0000 | $1.0000 \pm$ 0.0000 | $1.0000 \pm$ 0.0000 | $1.0000 \pm$ 0.0000 |
| AUROC↑ | $0.5000 \pm$ 0.0000 | $\mathbf{0.7750} \pm$ **0.0559** | $0.7500 \pm$ 0.0000 | $\mathbf{0.7750} \pm$ **0.0559** |

Table 7: Comparison of methods across metrics on Real-World Dataset scm_4.

The experimental results on the real-world dataset scm_4 are summarized in Table 7. Overall, all methods achieve the same SHD value, indicating limited differences in structural recovery. Nev-

ertheless, noticeable variations emerge in other metrics. Ours attains the highest MCC score, suggesting a relative advantage in overall prediction accuracy. Both iCCRL and our proposed method achieve the best AUROC, demonstrating superior robustness in distinguishing true from false causal edges. In contrast, the original CCRL records an AUROC of only 0.50, close to random guessing, which highlights its significant weakness on this dataset.

| Metric | CCRL | iCCRL | DCCRL | Ours |
|---|---|---|---|---|
| MCC↑ | $0.0188 \pm 0.0077$ | $0.0274 \pm 0.0061$ | $0.0249 \pm 0.0169$ | $\mathbf{0.0289} \pm \mathbf{0.0045}$ |
| SHD↓ | $\mathbf{0.2000} \pm \mathbf{0.4472}$ | $1.0000 \pm 0.0000$ | $1.0000 \pm 0.0000$ | $0.6000 \pm 0.5477$ |
| AUROC↑ | $\mathbf{1.0000} \pm \mathbf{0.0000}$ | $0.8000 \pm 0.0685$ | $0.7750 \pm 0.0559$ | $0.9250 \pm 0.0685$ |

Table 8: Comparison of methods across metrics on Real-World Dataset scm_5.

The experimental results on the real-world dataset scm_5 are presented in Table 8. In general, CCRL achieves an AUROC of 1.0, indicating excellent discriminative ability, but its MCC and SHD reveal limited success in recovering the causal structure. Our method achieves the highest MCC and a strong AUROC of 0.9250, demonstrating a balanced advantage in both accuracy and robustness. iCCRL achieves an AUROC of 0.80, slightly outperforming DCCRL, though both methods show limited structural recovery with SHD fixed at 1.0. In summary, while CCRL excels in edge discrimination, our proposed method provides the most balanced performance across all evaluation metrics.

In the three subsets of lt_camera_v1, the statistical properties of the noise shift from Gaussian to truncated Gaussian, which violates the assumption of additive Gaussian noise in the latent variable generation process of the baseline CCRL model. iCCRL, DCCRL, and our proposed method are all based on the CCRL model. As a result, the performance of all four methods is poor on the MCC metric.

### C.6.2 MERFISH BRAIN DATASET

Figure 12a shows the disentanglement performance on the MERFISH brain dataset using F1-score averaged over five random seeds. Our method achieves 0.193, 0.254, and 0.104 on Age, Domain, and Celltype factors, while CausCell achieves 0.155, 0.256, and 0.108. This shows our method's superior performance on the Age factor while maintaining comparable performance on the other two factors.

Furthermore, our method shows substantial advantages in matching score performance (Figure 12b), consistently outperforming CausCell across all five random seeds. This superior matching performance indicates that our method not only maintains competitive disentanglement capabilities but also significantly enhances the preservation of biologically relevant marker genes in the generated cells, thereby providing better support for downstream biological interpretation and analysis.

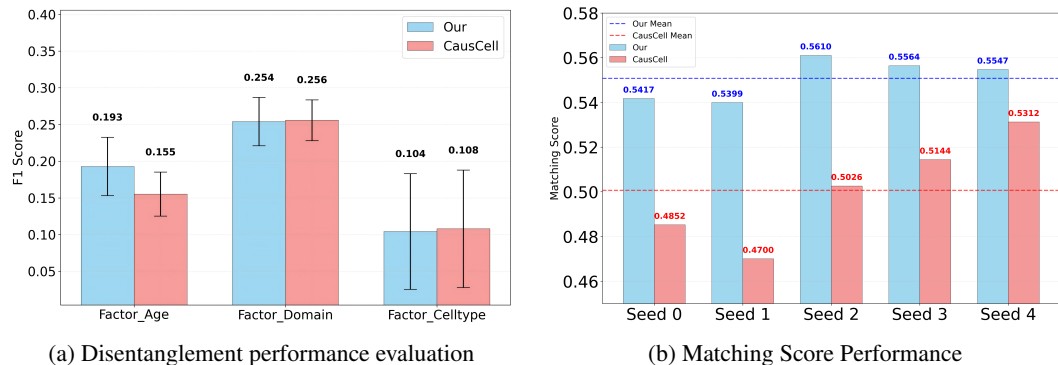

(a) Disentanglement performance evaluation      (b) Matching Score Performance

Figure 12: Performance comparison on MERFISH Brain dataset: (a) Disentanglement performance measured by F1-score, and (b) Matching score performance for marker gene preservation. Results show our method achieves comparable disentanglement while significantly improving matching scores compared to CausCell.

### C.6.3 LT_CRL_BENCHMARK_V1 DATASET

Figure 13a shows the evolution of three kernel bandwidths in our method: the bandwidth for observed embeddings $h_v$, the outer smoothing bandwidth $h_q$, and the particle-flow bandwidth $h_\rho$. During training, these bandwidths adapt automatically. Specifically, $h_v$ converges to a level that avoids over-smoothing the distribution of observed embeddings; $h_q$ adjusts to match the scale required by the score-mismatch term; and $h_\rho$ stabilizes the particle flow by suppressing spurious modes. All three bandwidths exhibit small fluctuations and remain relatively stable over training.

Figure 13b traces the per-dimension diagonal entries of the mobility matrix $\Sigma$ used in the weighted Wasserstein gradient flow. The five entries start at distinct scales; as training proceeds, each settles to a stable plateau.

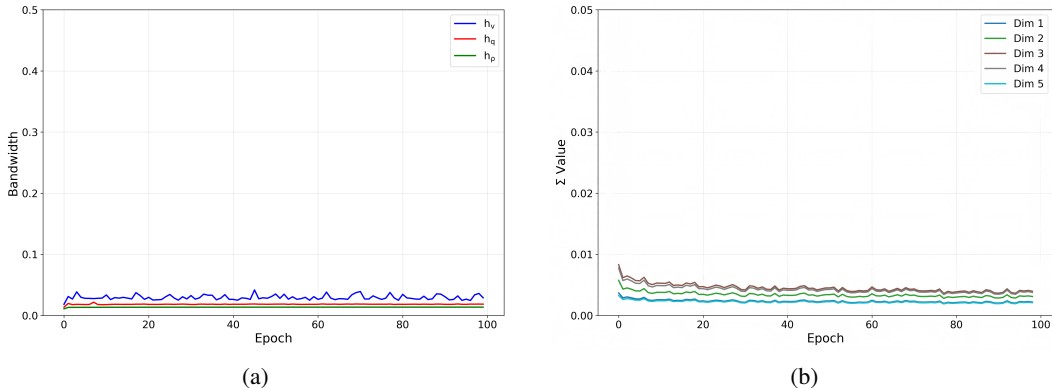

(a)                 (b)

Figure 13: Evolution of bandwidths and mobility on the lt_crl_benchmark_v1 dataset. (a) Adaptive kernel bandwidths during training. (b) Evolution of the five diagonal entries of the mobility matrix.

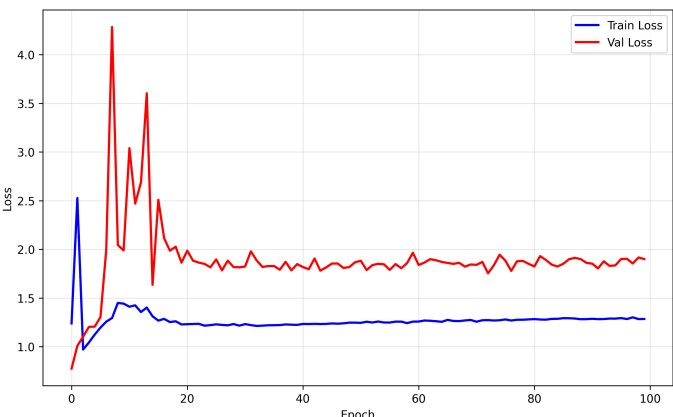

Figure 14: Training and validation losses curves on lt_crl_benchmark_v1 dataset.

Figure 14 shows the training and validation total loss curves on the lt_crl_benchmark_v1 dataset (seed = 0). Both losses are the sum of the main loss and the denoising loss. The training loss drops quickly in the early epochs and converges fast, while the validation loss fluctuates strongly at the beginning. As training proceeds, both curves gradually become stable.

## D USE OF LARGE LANGUAGE MODELS (LLMS)

In preparing this paper, we used a large language model (LLM, OpenAI's ChatGPT) as an auxiliary tool for language polishing, translation, rephrasing, and simplifying certain technical descriptions to improve readability and consistency. No LLM assistance in ideation, design, or results; all scientific contributions were done independently by the authors.

