# OpenReview forum: "Latent-Space Denoising for Causal Representation Learning via Free-Energy-Guided Wasserstein Particle Flows"
_ICLR.cc/2026/Conference — ICLR 2026 Conference Withdrawn Submission_

### Official Review · Reviewer_izqE · 2025-10-31

**Soundness:** 3
**Presentation:** 2
**Contribution:** 3
**Rating:** 8
**Confidence:** 3

**Summary:**

This paper proposes a latent-space denoising method for learning from data corrupted by unknown nonlinear mixing and noise. It uses an encoder to map observations into a low-dimensional latent space, where the corruption approximately becomes additive noise. The clean latent distribution is recovered by minimizing a free-energy objective:
$E(\rho) = D_{\mathrm{KL}} \left(q_{\rho} || \nu\right) + \lambda \mathrm{Ent}(\rho)$,
where $q_{\rho} = \rho * \varphi_{H}$ and $\varphi_{H}$ is the centered Gaussian kernel.
The authors derive the Wasserstein gradient flow of this objective, leading to a particle flow algorithm that
progressively denoises the latent representations.

The experiments show that this latent space denoiser can be used as a plug-in module for causal representation learning (CRL), improving the accuracy and stability of latent factor and causal structure recovery under realistic noise.

**Strengths:**

The paper presents a well-developed theoretical framework supported by convincing experiments with better performance compared to the baselines.

**Weaknesses:**

I am not very familiar with this field, and I found the paper quite difficult to follow. I struggled to understand several of the theoretical details and proofs, and I am still not confident that I fully grasp the details. I strongly recommend that the authors provide clearer explanations and more accessible notation to make the paper easier to follow, especially for readers who are not experts in this area.

**Questions:**

1. I am curious whether the proposed method can be extended to the case of indirect and noisy
observations of the clean signal $X$, in the context of ill-posed inverse problems.
Specifically, can the framework handle models of the form  $Y = A(X) + \epsilon$
where $A$ denotes a (possibly ill-posed) forward operator? This formulation is more general than
the additive-noise setup presented in the paper.

2. The choice of the latent dimension $d_u$ is not discussed in the experimental section.
Could the authors clarify how $d_u$ was selected for different experiments and elaborate on how the performance depends on this parameter?

---

> ### Author Response · Authors · 2025-11-21
>
> We appreciate the Reviewer izqE's constructive comments and helpful feedback. The issue you mentioned in the Weaknesses can be found in our response to Reviewer s7Xc, Question 1. If you have any further questions, please feel free to point them out. Below are our responses to the two questions you raised:
>
>
> > - **Q1:** I am curious whether the proposed method can be extended to the case of indirect and noisy observations of the clean signal X, in the context of ill-posed inverse problems. Specifically, can the framework handle models of the form $Y = A(X)+\epsilon$ where denotes a (possibly ill-posed) forward operator? This formulation is more general than the additive-noise setup presented in the paper.
>
> **A1:** According to the definition of ill-posed problems in [1], an ill-posed inverse problem arises when the forward operator $A$ is non-invertible. In this case, although the mapping $f$ is a bijection from $\mathbb{R}^{d_u}$ to $f(\mathbb{R}^{d_u})$, the operator $A$ does not satisfy the bijective condition from $f(\mathbb{R}^{d_x})$ to $A \circ f(\mathbb{R}^{d_x})$. To address this issue, we consider the composite mapping $A \circ f$. We notice that when $f(z_1) - f(z_2) \notin \ker(A)$ for any $z_1 \ne z_2$, the mapping $A \circ f: \mathbb{R}^{d_u} \rightarrow A \circ f(\mathbb{R}^{d_u})$ becomes injective and further bijective, thereby meeting the requirement of an invertible mapping. Once this composite mapping is invertible, our method can be applied to solve such inverse problems.
>
> > - **Q2:** The choice of the latent dimension $d_u$ is not discussed in the experimental section. Could the authors clarify how $d_u$ was selected for different experiments and elaborate on how the performance depends on this parameter?
>
>
> **A2:** As stated in lines 191–192 of the paper, in different experimental settings $d_u$ should satisfy $d_x \ge d_u \ge d_z$, where $d_x$ is the dimension of the observation space, $d_u$ is the dimension of the low-dimensional latent space, and $d_z$ is the dimension of the latent variables. In many cases, we have some prior knowledge about $d_z$; when $d_z$ is known, we typically set $d_u = d_z$. However, when $d_z$ is unknown, our experimental results show that smaller values of $d_u$ tend to give better performance.
>
> |    ours    | TPR ↑ | FPR ↓ | FDR ↓ | AUROC ↑ |
> |:----------:|:----:|:----:|:----:|:-----:|
> | lat_dim=6|0.720±0.228|0.024±0.036|0.097±0.136|0.854±0.108|
> | lat_dim=8|0.640±0.261|0.011±0.016|0.164±0.157|      0.815±0.126|
> | lat_dim=10|0.600±0.141|0.005±0.006|0.090±0.125|  0.798±0.070|
>
>
>
> We sincerely thank you for your constructive feedback, the time you have dedicated, and your recognition of our work. We especially appreciate the valuable suggestions regarding future directions, which we find highly inspiring and will consider in our subsequent research.
>
> [1] Kirsch, Andreas, et al. "An introduction to the mathematical theory of inverse problems." SIAM Review. 1997.

---

### Official Review · Reviewer_4hGD · 2025-11-01

**Soundness:** 2
**Presentation:** 2
**Contribution:** 1
**Rating:** 2
**Confidence:** 4

**Summary:**

The authors attempt to recover latent variable from observation to obtain a causal representation learning framework. The main approach is to model $Y=f(Z) + \epsilon$, and the first part is to say that X=f(Z) is identifiable and then will use X to recover Z. Experiments on several datasets is ok.

However, if my understanding is correct, the proof of identifiability of X is wrong. The basic idea of the proof is to use the fact that the characteristic function of Y equals the multiplication of characteristic of X and $\epsilon$ and to say that the distribution of X is solely determined by the distribution of $Y$. However, first if we do not assumption the distribution of $\epsilon$, we cannot get the distribution of $X$. Secondly, the derivation in line 658 is highly like not correct. What do you mean by law of Y? If 658 holds it may be a contradiction with the fact that $X$ and $\epsilon$ are independent.

Another problem is to recover Z from X, generally, without further assumptions, just with the condition that $f$ is injective, this would not be possible.

**Strengths:**

As the basic framework of the paper do have some technical issues, the only strength comes from the experimental part.

**Weaknesses:**

1. The proof if proposition is highly likely to be wrong.

2. Recover Z from X is in generally not possible.

**Questions:**

See my previous comments about the technique issues.

---

> ### Author Response · Authors · 2025-11-15
>
> Dear Reviewer 4hGD, thank you for your thought. Your feedback has improved the clarity of our manuscript, and we sincerely appreciate the time and effort you dedicated to reviewing our work.
>
> > - **Summary1:** The authors attempt to recover latent variable from observation to obtain a causal representation learning framework. The main approach is to model $Y=f(Z)+\epsilon$, and the first part is to say that $X=f(Z)$ is identifiable and then will use $X$ to recover $Z$. Experiments on several datasets is ok.
> >However, if my understanding is correct, the proof of identifiability of $X$ is wrong. The basic idea of the proof is to use the fact that the characteristic function of $Y$ equals the multiplication of characteristic of $X$ and $\epsilon$ and to say that the distribution of $X$ is solely determined by the distribution of $Y$. However, first if we do not assumption the distribution of $\epsilon$, we cannot get the distribution of $X$. Secondly, the derivation in line 658 is highly like not correct. What do you mean by law of $Y$? If 658 holds it may be a contradiction with the fact that $X$ and $\epsilon$ are independent.
> > - **W1:** The proof if proposition is highly likely to be wrong.
>
>
> **A1:** We sincerely thank you for your examination of this derivation. There may have been misunderstandings or confusion regarding the concept and proof. Below are point-by-point explanations.
>
> - **Clarification on Identifiability in Proposition 1**
> To avoid conceptual confusion, we emphasize that the notion of identifiability in Proposition 1 is established through distributional equivalence:$$\mathrm{Law}(Y_1) = \mathrm{Law}(Y_2) \implies \mathrm{Law}(X_1) = \mathrm{Law}(X_2).$$ Proposition 1 shows that in an additive model with a fixed noise if two observed variables $Y_1$ and $Y_2$ share the same distribution, then the corresponding signals $X_1$ and $X_2$ must also have the same distribution. Notice that the identifiability of parameters is defined in a similar way [1].
>
> - **The proof for Proposition 1**
> In Proposition 1, we consider the case $$Y_1 = X_1 + \varepsilon,  Y_2 = X_2 + \varepsilon; X_i \\perp\\!\\!\\!\\perp \varepsilon,i=1,2.$$ Using independence and standard properties of characteristic functions, the corresponding $X_1$ and $X_2$ have the same probability distribution when $Y_1$ and $Y_2$ share the same probability distribution. Our proof from Line 660 to 665, which is based on characteristic functions, is inspired by the work presented in Appendix A.2, Theorem 4 of Lachapelle et al. [1].
>
>     We are sorry for your misunderstanding on the identifiability and the proof for Proposition 1. We will try to make our Proposition 1 more explicit in the revised manuscript to avoid ambiguity.
>
> - **Signal recovery when the noise distribution is unknown.**
> In practice, many denoising methods can operate under unknown noise distributions. Take a toy additive model example: $$Y = \mu + \varepsilon, \mu \\perp\\!\\!\\!\\perp \varepsilon,$$ where $\mu$ is a true signal and $\varepsilon$ has an unknown distribution with zero mean. So we have $\mathbb{E}[Y]=\mu.$ Thus, the sample mean $\bar{Y}$ serves as the moment-based estimator $\hat{\mu}$ of $\mu$. This demonstrates that we can still estimate the signal $\mu$ even when the noise distribution is unknown.
>
> - **Clarification on "$\mathrm{Law}(Y)$"**
> As described in from Line 182 to 184, "$\mathrm{Law}(Y)$" denotes the probability law of $Y.$ It means that: $$\mathrm{Law}(Y)(A) = \mathbb{P}(Y \in A), \quad \forall A \in \mathcal{B}(\mathbb{R}^{d_u}).$$ Thus, “$\mathrm{Law}(Y_1) = \mathrm{Law}(Y_2)$” simply means the two random variables share the same distribution. Thank you for pointing out that. We will add this description in the proof of Proposition 1 in the Appendix A.1.
>
>
> > - **Summary2:** Another problem is to recover $Z$ from $X$, generally, without further assumptions, just with the condition that $f$ is injective, this would not be possible.
> > - **W2:** Recover Z from X is in generally not possible.
>
> **A2:** Your point is entirely correct: the mere assumption that $f$ is injective is not sufficient to recover $Z$ from $X$. We would like to clarify that the additional assumptions involved are those specified in the classical CCRL framework [2]. We did not elaborate in detail on those additional assumptions, which are not the primary focus of our work, since our denoising method can work as a drop-in module for the CCRL. We are considering adding more descriptions on the identifiability in the revised version.
>
>
> [1] Lachapelle, Sébastien, et al. "Disentanglement via mechanism sparsity regularization: A new principle for nonlinear ICA." Conference on Causal Learning and Reasoning. 2022.
>
> [2] Buchholz, Simon, et al. "Learning linear causal representations from interventions under general nonlinear mixing." Advances in Neural Information Processing Systems. 2023.

---

### Official Review · Reviewer_s7Xc · 2025-11-04

**Soundness:** 3
**Presentation:** 1
**Contribution:** 2
**Rating:** 4
**Confidence:** 2

**Summary:**

The authors propose a technique to adapt existing CRL methods to real-world observations that are corrupted. They do this by latent space denoising, where the embeddings of corrupted data are mapped to behave approximately as additive noise. Then recover a clean latent distribution via free-energy minimisation, combining KL divergence and entropy regularisation for stability. Authors demonstrate that the resulting latent-space denoiser can be seamlessly integrated into existing CRL pipelines, yielding substantial gains in accuracy and causal structure recovery on both real-world and simulated noisy datasets.

**Strengths:**

- This paper deals with a practical problem of applying CRL methods to real-world noise data

- The idea of performing denoising in the latent space using free-energy minimisation and Wasserstein particle flows is novel

- The authors refer to many works in the field, in-depth literature review

**Weaknesses:**

- Hard to follow the work, the content is very dense in some parts of the paper; easing it out would help the readers

-  The motivation to link Wasserstein denoising and causal mechanism recovery is mostly hand-wavy

- The nonvanishing characteristic functions argument for identifiability is interesting, while the paper doesn't discuss the limitations of estimation methods

- In lines 193-200 authors discuss how a trained encoder will result in an additive noise scenario, but the constraints required to achieve that are not that clear

- It would be really helpful to provide intuition for your theorem

**Questions:**

See weakness section

---

> ### Author Response · Authors · 2025-11-20
>
> Dear Reviewer s7Xc, thank you for your careful review of our work and for your constructive comments. We greatly appreciate the time and effort you have devoted to the review. Below we provide point-by-point responses to your questions.
>
> **A1(W1)** Thanks for your valuable comments and suggestions. To improve the readability of the paper, we plan to revise the content related to the Wasserstein gradient flow method in Section 3.3 in our revised version as follows:
> > We present the basic definitions of the $\Sigma$-weighted Wasserstein metric on the space of probability densities, together with the induced tangent space, $\Sigma$-weighted norm, and $\Sigma$-weighted inner product.
> > Let $\Sigma:\Omega\to\mathbb{R}^{d_u\times d_u}$ be a smooth, symmetric, positive-definite mobility. For $\rho\in\mathcal P_2(\Omega)$, define the $\Sigma$-weighted velocity space $$L^2_{\rho,\Sigma^{-1}}(\Omega;\mathbb{R}^{d_u})=\\{w:\int_\Omega w^{\top}\Sigma^{-1}w\rho du<\infty\\}$$ and the pre-tangent set $$T_\rho\mathcal P_2(\Omega)=\\{-\nabla\cdot(\rho w):w\in L^2_{\rho,\Sigma^{-1}}\\}.$$ For $\sigma=-\nabla\cdot(\rho w)$ we define the $\Sigma$-weighted norm as $$\||\sigma\||^2_{\rho,\Sigma}=\int_\Omega\nabla\phi^{\top}\Sigma\nabla\phi\rho du,$$ yielding a local $\Sigma$-weighted Wasserstein (Riemannian) structure on $\mathcal P_2(\Omega)$.For $\sigma_i=-\nabla\cdot(\rho\Sigma\nabla\phi_i)$ we define the $\Sigma$-weighted inner product as $$\langle\sigma_1,\sigma_2\rangle_{\rho,\Sigma}=\int_\Omega (\nabla\phi_1)^{\top}\Sigma\nabla\phi_2\rho du.$$ Further details can be found in Appendices A.3 and A.4.
>
> In addition, we will add the intuition of Theorem 1 from Question 5 to the main text. The content is as follows:
> >  Similar to the optimization in finite-dimensional spaces, Theorem 1 provides a characterization of steepest descent and energy dissipation in the Riemannian space. The latent density function $\rho$ can be updated in the Riemannian space along the direction of the steepest descent of the energy function $\mathcal{E}(\rho)$.
>
> We hope that these revisions alleviate the issue of overly dense content and improve the readability of the paper. If you have any further questions or concerns about the paper, please let us know. We look forward to receiving your continued valuable feedback.
>
> **A2(W2)** Causal representation learning is usually performed in the latent space, thus the density of the latent variables needs to be estimated in this space. Our method estimates the latent density $\rho$ in a Riemannian space. When the distribution of the latent variables is affected by noise, Riemannian geometric tools are required for denoising. To address this, we incorporate the denoising mechanism into the objective function defined in the Riemannian space. By optimizing this objective, we perform denoising and density estimation in a unified way.
>
> We plan to add the research motivation to the main text in a future version to further improve the overall readability of the paper.
>
> **A3(W3)** Thanks for your comments. Our Proposition 1 holds given a noise with an infinitely divisible distribution. However, in some cases, the noise may not meet this assumption on infinitely divisible distribution. Relevant discussions can be found in Appendix A.1 of the paper. In addition, we adopt a nonparametric estimation method, as noted in the Limitation section of the paper: “While this nonparametric choice is flexible, it remains sensitive to dimensionality and sample size, reflecting a bias-variance trade-off.”
>
> **A4(W4)** This section provides an intuitive explanation for why the embeddings exhibit a certain degree of additivity. We account for this additivity by applying a first-order Taylor expansion to the encoder $h$. This approach requires differentiability conditions on $h$. When $h$ is infinitely differentiable, the Taylor expansion can accurately capture the additivity, as long as the noise remains within the radius of convergence at $X$. When $h$ is first-order differentiable, we give an approximation that is valid  when the noise level is sufficiently small. Finally, under the local “frozen Jacobian’’ assumption, the embedding distribution can be approximated as the convolution of the clean embedding distribution with a noise term induced by the local linearization.
>
> **A5(W5)** Thank you for your comment. Similar to the optimization in finite-dimensional spaces, Theorem 1 provides a characterization of steepest descent and energy dissipation in the Riemannian space. Figure 3 gives a more intuitive illustration, showing the geometric meaning of the Wasserstein Riemannian gradient. The latent density $\rho$ can be updated in the Riemannian space along the direction of the steepest descent of the energy function $\mathcal{E}(\rho)$.
>
> Once again, we thank you for your valuable feedback and for your interest in our work. Your comments have been very helpful in improving the quality and readability of the paper.

---

### Official Review · Reviewer_VWkd · 2025-11-04

**Soundness:** 2
**Presentation:** 3
**Contribution:** 2
**Rating:** 4
**Confidence:** 3

**Summary:**

This paper addresses the challenge of causal representation learning (CRL) from observations that are corrupted by both unknown nonlinear mixing and additive noise. The authors propose a new method called Latent-Space Denoising, which acts as a "plug-and-play" module upstream of an existing CRL model. The core idea is to first learn an encoder $h$ that maps noisy observations $Y = X + \epsilon$ (where $X=f(Z)$ is the nonlinearly mixed clean data) into a latent space $U= h(Y)$. The authors justify that, via a first-order Taylor expansion, the latent representation of the noisy data can be approximated as the latent representation of the clean data plus an additive noise term: $h(Y) \approx h(X) + J_h(X)\epsilon$. Given this latent additive noise model, the paper proposes to recover the "clean" latent distribution $p(Z)$ from the "noisy" latent distribution $p(U)$ by minimizing a free-energy objective. This optimization is implemented using a free-energy-guided Wasserstein particle flow. The resulting denoised latent representations are then fed into a standard CRL method (in this case, CCRL) to identify the underlying causal system. Experiments on synthetic and semi-synthetic benchmarks show that adding the Latent-Space Denoising module improves the performance.

**Strengths:**

Problem Significance: The paper addresses the critical and practical limitation of CRL methods, which often assumes the observations are the noiseless mixing of latent causal variables.

Originality: The proposed  Latent-Space Denoising method, which combines a learned encoder with a free-energy-guided particle flow for denoising in the latent space to help causal identification, is a novel and interesting approach.

Methodological Novelty: The application of Wasserstein particle flows to recover a clean latent distribution for a downstream discriminative task (CRL) looks like a technically non-trivial contribution.

**Weaknesses:**

Weak Justification for Latent Additivity: The paper's primary assumption that additive noise in the observation space translates to an approximate additive noise model in the latent space based on a first-order Taylor approximation. This argument is not convincing to me, as it seems to rely on the component wise identifiability of the inverse of the ground truth mixing map. Even in the noise-free cases, oftentimes the component wise identifiability cannot be guaranteed (only group wise). Even under strong conditions where the component wise identifiability can be achieved, with the additional noise in the observation, the identifiability does not automatically hold. Plus, when the noise is large the first-order Taylor approximation may not perform well.  Thus, the validity of this approximation, which is essential to the method, is not sufficiently investigated and justified.

Identifiability from Noisy Data: The paper does not theoretically address how its two-stage process (denoising then identification) impacts the identifiability guarantees of the baseline CRL model. Standard CRL identifiability results are for noise-free data. The authors need to provide justification that their denoising step successfully recovers a representation from which the true causal factors are still identifiable.

Limited "Plug-and-play" Validation: The paper claims LSD is a "plug-and-play" module, but this is only demonstrated by integrating it with CCRL. To substantiate this claim, the authors should show results with other, diverse CRL methods (e.g., VAE-based or other contrastive methods) to prove its general applicability. Plus, it would help to also compare against the most obvious and direct baseline: denoising in the observation space and then applying the CRL algorithm.

**Questions:**

1. Could the authors provide more empirical and theoretical evidence for the validity of the first-order Taylor approximation?

2. Why did the authors not compare against the more direct baseline of (1) training a denoising model (e.g., DAE) in the observation space and (2) applying the CCRL model?

3. How does the proposed method provably ensure that the introduced additional steps preserves the necessary identifiability conditions for a baseline CRL model (e.g. CCRL)?

4. To better support the "plug-and-play" claim, could the authors provide results from integrating the method with at least one other CRL baseline?

---

> ### Author Response · Authors · 2025-11-21
>
> We appreciate the reviewer VWkd's constructive comments and helpful feedback.
>
> **A1(Q1):** Thanks for your question and for your interest in the identifiability of our method. We provide a unified clarification regarding identifiability and the role of the first-order Taylor expansion.
>
> First, we need to clarify a misunderstanding. The first-order Taylor expansion in the paper is not used as the theoretical basis for identifiability. Its purpose is to explain why the embeddings show approximate additivity, which helps motivate the algorithmic design. In other words, the Taylor expansion illustrates the local additivity of the encoder $h$; it is not part of the identifiability argument.
>
> The identifiability guarantees come from two sources:
>
> 1. **Proposition 1:** With a given infinitely divisible noise, the distribution of $X$ is uniquely determined by the distribution of $Y$, ensuring identifiability from noisy to clean observations.
> 2. **The CCRL framework:** After denoising, the identifiability of the latent variables $Z$ from $X$ is ensured by classical CCRL[1] theory, which has been rigorously established in prior work.
>
> Thus, the denoising module can not change the identifiability of CCRL . A more detailed discussion of identifiability will be included in a future version of the paper.
>
> Finally, we note that prior works on component-wise identifiability (e.g., [2], [3], [4]) typically rely on the uniqueness of spectral decompositions or sparsity assumptions, whereas CCRL relies on interventional information. These approaches follow different theoretical paths and apply under different conditions.
>
> **A2(Q2):** Thank you for your question. We have actually included this comparison in our original experiments. Specifically, iVAE and DVAE are denoising autoencoders trained in the observation space, and then the CCRL model is applied to the denoised observations. The detailed implementation is provided in the Figure 11 of Appendix C.4. The experimental results in Sections 4.5.1 and 4.5.2 show that denoising in the latent space achieves better performance than denoising in the observation space, which provides the validation of our method's design choice.
>
> **A3(Q3):** According to Proposition 1, our method can recover the clean observations. These denoised observations are then fed into the CCRL framework to achieve identifiability of the latent distribution. For a detailed discussion, please refer to our response to Question 1.It is important to emphasize that the introduced denoising module does not alter the identifiability conditions of CCRL and does not compromise its theoretical guarantees. Therefore, from a theoretical standpoint, the added denoising step preserves the identifiability assumptions required by CCRL.
>
> **A4(Q4):** Thank you for your question. Regarding the plug-and-play experimental validation, we have integrated our method into causal disentanglement for single-cell representations[5] in Section 4.5.2, with detailed results provided in Table 4 and Appendix C.6.2.
>
> To further verify the generalizability of our method, following your suggestion, we integrated our method with the VAE-based method – CausalVAE[6]. We used its flow dataset and added Gaussian noise with std=30 and std=50. The experimental results on five random seeds are as follows:
>
> |std=30|MCC↑|SHD↓|AUROC↑|
> |:-:|:-:|:-:|:-:|
> |CausalVAE|0.8731±0.0030|1.80±0.40|0.7222±0.0378|
> |Ours|0.8830±0.0048|1.60±0.49|0.8111±0.0363|
>
> |std=50|MCC↑|SHD↓|AUROC↑|
> |:-:|:-:|:-:|:-:|
> |CausalVAE|0.7175±0.0050|2.00±0.63|0.7593±0.0895|
> |Ours|0.7252±0.0040|2.20±0.40|0.8000±0.1564|
>
>
> These results show that our method can be effectively integrated with different CRL frameworks, supporting the plug-and-play claim.
>
> We sincerely thank you for your constructive feedback, the time you have dedicated, and your recognition of our work. We especially appreciate the valuable suggestions regarding future directions, which we find highly inspiring and will consider in our subsequent research.
>
> [1] Buchholz, Simon, et al. "Learning linear causal representations from interventions under general nonlinear mixing." Advances in Neural Information Processing Systems. 2023.
>
> [2]Li, Zijian, et al. "Towards Identifiability of Hierarchical Temporal Causal Representation Learning." arXiv preprint arXiv:2510.18310(2025).
>
> [3]Fu, Minghao, et al. "Learning general causal structures with hidden dynamic process for climate analysis." arXiv preprint arXiv:2501.12500v2(2025).
>
> [4]Li, Zijian, et al. "Online Time Series Forecasting with Theoretical Guarantees." arXiv preprint arXiv:2510.18281(2025).
>
> [5]Gao, Yicheng, et al. "Causal disentanglement for single-cell representations and controllable counterfactual generation." Nature communications. 2025.
>
> [6]Yang, Mengyue, et al. "Causalvae: Disentangled representation learning via neural structural causal models." Proceedings of the IEEE/CVF conference on computer vision and pattern recognition. 2021.

---

### Author Response · Authors · 2025-12-03

Dear Area Chair,

We sincerely thank you for dedicating your valuable time and effort to review our work under unexpected circumstances. Since we can no longer communicate effectively with the reviewers, we believe it is necessary to directly outline the key contributions of our study and point out several serious issues that arose during the review process.

Our main contribution is to address the problem that existing causal representation learning methods often fail when real observational data contain noise [1]. We tackle this challenge by deriving a **weighted Wasserstein gradient** on a Riemannian geometric space and designing an **explicit particle flow algorithm** to perform denoising in the latent space.

We greatly appreciate the positive comments provided by the reviewers on our work, for example:

- “a novel and interesting approach,”
- “looks like a technically non-trivial contribution,” and
- “presents a well-developed theoretical framework supported by convincing experiments with better performance compared to the baselines.”


However, it is very regrettable that Reviewers VWkd, s7Xc, and 4hGD did not correctly understand our method and even showed serious misunderstandings of the basic ideas behind our model. Specifically:

**Reviewer VWkd** argued that using a first-order Taylor approximation to construct an additive noise model in the latent space does not effectively guarantee identifiability.

- We have already explained that the identifiability of our method is ensured by Proposition 1 in the paper and by the identifiability of the CCRL [2] framework. The role of the first-order Taylor approximation is simply to explain why the embeddings exhibit approximate additivity, thereby providing motivation for the algorithmic design.

**Reviewer s7Xc** considered our content difficult to read and did not understand the relationship between our Wasserstein denoising procedure and causal mechanism recovery.

- We pointed out that causal representation learning is typically performed in the latent space, and therefore it is necessary to estimate the density of the latent variables in that space. Our method estimates the latent density on a Riemannian space. When the distribution of the latent variables is corrupted by noise, Riemannian geometric tools are required for denoising.

**Reviewer 4hGD** argued that the basic framework of our method is flawed, that the identifiability proof is incorrect, and even that recovering latent variables from observed variables is impossible.

- We have also provided detailed responses to Reviewer 4hGD. However, we must note that Reviewer 4hGD’s complete rejection of our work was made without an understanding of the basic background of causal representation learning. We respectfully ask you to take this into account in the final evaluation.

Given the originality and significance of our method, as well as the empirical evidence we have presented, we believe our work will provide valuable insights to the ICLR community. We sincerely ask you to carefully weigh these factors in your final decision.


[1] Gamella, Juan L, et al. "Sanity checking causal representation learning on a simple real-world system." International conference on machine learning. 2025.

[2] Buchholz, Simon, et al. "Learning linear causal representations from interventions under general nonlinear mixing." Advances in Neural Information Processing Systems. 2023.

Sincerely,

Authors

---

### Note · Authors · 2026-01-31

**Comment:**

We have decided to withdraw our paper, "Latent-Space Denoising for Causal Representation Learning via Free-Energy-Guided Wasserstein Particle Flows," from ICLR 2026. The primary reason for this decision is that our paper was rejected by ICLR 2026, and we plan to submit the paper to another journal. We wish to avoid any potential overlap or conflicts due to the ongoing discussion and feedback in the ICLR submission process.

**Withdrawal Confirmation:**

I have read and agree with the venue's withdrawal policy on behalf of myself and my co-authors.

---

### Meta-Review · Area_Chair_Z1ng · 2025-12-19

**Summary:**

This paper looks at the problem of learning causal representations with noise. Due to the unique situation, I have carefully read the paper myself before looking at the reviewer’s feedback in detail.


- Unfortunately, I find the premise of the paper incorrect. They cite Gamella et al. as motivation, which has several serious experimental flaws. In fact, the paper only considers weak baselines (largely CCRL and variants based on iVAE), which was also raised by reviewers. Theoretically, one can model the noise as another factor of variation, that is independent from the others and thus in principle even identifiable.

- As mentioned above, the paper uses weak baselines and insufficient evaluation metrics (missing R2). I am particularly concerned about the evaluation, as all these metrics are known to have issues: it’s easy to construct counterexamples where MCC/SHD (and R2) are high despite entanglement.

- The authors should also check the performance of a supervised baseline on their data, as the data is known to have issues.

I agree with the authors that the concern on identifiability of VWkd is fine. Yet, it could be interesting for the authors to expand on the point that one could (or perhaps not in this setting) treat the noise like any other latent variable that can be identified. I think that would make the paper a lot stronger, showing theoretically that the additive noise would lead to non-identifiability. This could also better address the question of 4hGD.

Generally the writing can also be streamlined a bit, although I found the paper well polished.

**Reviewer Concerns:**

The identifiability concerns were not fully addressed by the authors. I also have questions about the strength of the baselines and choice of the data set. The baselines were also criticised by the reviewers.

**Reviewer Scores:**

I don't think they would have changed their scores. I also would have asked them about these issues and likely have rejected the paper anyway.

---

### Decision · Program_Chairs · 2026-01-26

Reject